# Air quality improvement and cognitive decline in community-dwelling older women in the United States: A longitudinal cohort study

Diana Younan[1☯]*, Xinhui Wang[2☯], Joshua Millstein[1], Andrew J. Petkus[2], Daniel P. Beavers[3], Mark A. Espeland[3], Helena C. Chui[2], Susan M. Resnick[4], Margaret Gatz[5], Joel D. Kaufman[6], Gregory A. Wellenius[7], Eric A. Whitsel[8], JoAnn E. Manson[9], Stephen R. Rapp[10], Jiu-Chiuan Chen[1,2]

1 Department of Population and Public Health Sciences, University of Southern California, Los Angeles, California, United States of America, 2 Department of Neurology, University of Southern California, Los Angeles, California, United States of America, 3 Department of Biostatistics and Data Science, Wake Forest School of Medicine, Winston-Salem, North Carolina, United States of America, 4 Laboratory of Behavioral Neuroscience, National Institute on Aging, Baltimore, Maryland, United States of America, 5 Center for Economic and Social Research, University of Southern California, Los Angeles, California, United States of America, 6 Departments of Environmental & Occupational Health Sciences, Medicine, and Epidemiology, University of Washington, Seattle, Washington, United States of America, 7 Department of Environmental Health, Boston University, Boston, Massachusetts, United States of America, 8 Departments of Epidemiology and Medicine, University of North Carolina, Chapel Hill, North Carolina, United States of America, 9 Department of Medicine, Brigham and Women's Hospital, Harvard Medical School, Boston, Massachusetts, United States of America, 10 Departments of Psychiatry and Behavioral Medicine and Social Sciences and Health Policy, Wake Forest School of Medicine, Winston-Salem, North Carolina, United States of America

☯ These authors contributed equally to this work.
* dyounan@usc.edu

**Data Availability Statement:** Data, codebook, and analytic code used in this report are held by the NIH-funded Coordinating Center of the Women's

## Abstract

### Background

Late-life exposure to ambient air pollution is a modifiable risk factor for dementia, but epidemiological studies have shown inconsistent evidence for cognitive decline. Air quality (AQ) improvement has been associated with improved cardiopulmonary health and decreased mortality, but to the best of our knowledge, no studies have examined the association with cognitive function. We examined whether AQ improvement was associated with slower rate of cognitive decline in older women aged 74 to 92 years.

### Methods and findings

We studied a cohort of 2,232 women residing in the 48 contiguous US states that were recruited from more than 40 study sites located in 24 states and Washington, DC from the Women's Health Initiative (WHI) Memory Study (WHIMS)-Epidemiology of Cognitive Health Outcomes (WHIMS-ECHO) study. They were predominantly non-Hispanic White women and were dementia free at baseline in 2008 to 2012. Measures of annual (2008 to 2018) cognitive function included the modified Telephone Interview for Cognitive Status (TICSm) and the telephone-based California Verbal Learning Test (CVLT). We used regionalized universal kriging models to estimate annual concentrations (1996 to 2012) of fine particulate

Health Initiative at the Fred Hutchinson Cancer Research Center and may be accessed as described on the Women's Health Initiative website: https://www.whi.org/page/working-with-whi-data.

**Funding:** This study is supported by the National Institute of Environmental Health Sciences (R01ES025888; J-C.C. and J.D.K.; and 5P30ES007048), the National Institute on Aging (NIA) (RF1AG054068; J-C.C.), and the Alzheimer's Disease Research Center at USC (NIA; P50AG005142 and P30AG066530). D.Y. and J-C.C. are supported in part by the NIA (P01AG055367). D.Y. is also supported by a grant from the Alzheimer's Association (AARF-19-591356). The air pollution models were developed under a STAR research assistance agreement, No. RD831697 (MESA Air) and RD-83830001 (MESA Air Next Stage), awarded by the US Environmental Protection Agency (EPA). M.A.E. receives funding from the Wake Forest Alzheimer's Disease Core Center (P30AG049638–01A1). S.M.R. is supported by the Intramural Research Program, NIA, NIH. The WHI program is funded by the National Heart, Lung, and Blood Institute, National Institutes of Health, U.S. Department of Health and Human Services through contracts HHSN268201600018C, HHSN268201600001C, HHSN268201600002C, HHSN268201600003C, and HHSN268201600004C. A list of contributors to WHI is available at https://www-whi-org.s3.us-west-2.amazonaws.com/wp-content/uploads/WHI-Investigator-Long-List.pdf. The funders had no role in study design, data collection and analysis, decision to publish, or preparation of the manuscript.

**Competing interests:** The authors have declared that no competing interests exist.

**Abbreviations:** AD, Alzheimer disease; APoE, Apolipoprotein E; AQ, air quality; BMI, body mass index; CVD, cardiovascular disease; CVLT, California Verbal Learning Test; EPA, Environmental Protection Agency; IQR, interquartile range; $NO_2$, nitrogen dioxide; $PM_{2.5}$, fine particulate matter; ppb, parts per billion; STROBE, Strengthening the Reporting of Observational Studies in Epidemiology; TICSm, modified Telephone Interview for Cognitive Status; WHI, Women's Health Initiative; WHI-HT, Women's Health Initiative hormone therapy; WHIMS, Women's Health Initiative Memory Study; WHIMS-ECHO, Women's Health Initiative Memory Study-Epidemiology of Cognitive Health Outcomes.

matter ($PM_{2.5}$) and nitrogen dioxide ($NO_2$) at residential locations. Estimates were aggregated to the 3-year average immediately preceding (recent exposure) and 10 years prior to (remote exposure) WHIMS-ECHO enrollment. Individual-level improved AQ was calculated as the reduction from remote to recent exposures. Linear mixed effect models were used to examine the associations between improved AQ and the rates of cognitive declines in TICSm and CVLT trajectories, adjusting for sociodemographic (age; geographic region; race/ethnicity; education; income; and employment), lifestyle (physical activity; smoking; and alcohol), and clinical characteristics (prior hormone use; hormone therapy assignment; depression; cardiovascular disease (CVD); hypercholesterolemia; hypertension; diabetes; and body mass index [BMI]). For both $PM_{2.5}$ and $NO_2$, AQ improved significantly over the 10 years before WHIMS-ECHO enrollment. During a median of 6.2 (interquartile range [IQR] = 5.0) years of follow-up, declines in both general cognitive status (β = −0.42/year, 95% CI: −0.44, −0.40) and episodic memory (β = −0.59/year, 95% CI: −0.64, −0.54) were observed. Greater AQ improvement was associated with slower decline in TICSm ($β_{PM2.5improvement}$ = 0.026 per year for improved $PM_{2.5}$ by each IQR = 1.79 μg/m$^3$ reduction, 95% CI: 0.001, 0.05; $β_{NO2improvement}$ = 0.034 per year for improved $NO_2$ by each IQR = 3.92 parts per billion [ppb] reduction, 95% CI: 0.01, 0.06) and CVLT ($β_{PM2.5\ improvement}$ = 0.070 per year for improved $PM_{2.5}$ by each IQR = 1.79 μg/m$^3$ reduction, 95% CI: 0.02, 0.12; $β_{NO2improvement}$ = 0.060 per year for improved $NO_2$ by each IQR = 3.97 ppb reduction, 95% CI: 0.005, 0.12) after adjusting for covariates. The respective associations with TICSm and CVLT were equivalent to the slower decline rate found with 0.9 to 1.2 and 1.4 to 1.6 years of younger age and did not significantly differ by age, region, education, Apolipoprotein E (ApoE) e4 genotypes, or cardiovascular risk factors. The main limitations of this study include measurement error in exposure estimates, potential unmeasured confounding, and limited generalizability.

## Conclusions

In this study, we found that greater improvement in long-term AQ in late life was associated with slower cognitive declines in older women. This novel observation strengthens the epidemiologic evidence of an association between air pollution and cognitive aging.

## Author summary

### Why was this study done?

- Although studies have shown that late-life exposure to outdoor air pollution is a modifiable risk factor for dementia, the epidemiological evidence for cognitive decline has been inconsistent.

- Epidemiological studies have demonstrated that improved air quality (AQ) may decrease mortality and improve respiratory health, strengthening the evidence of a relationship between ambient air pollution and these health outcomes.

- To our knowledge, no previous studies have examined the potential benefit of slowing cognitive aging by AQ improvement.

## What did the researchers do and find?

- Using a US cohort of 2,232 older women followed up to 20 years, we explored whether women living in locations with greater AQ improvement had slower decline in their cognitive function.

- AQ improvement was defined as the difference in air pollution levels over 2 time points that were 10 years apart.

- Living in locations with greater AQ improvement was associated with slower cognitive declines in older women, equivalent to women who were 0.9 to 1.6 years younger.

- The associations were similar across age groups, geographic region, education, Apolipoprotein E (ApoE) e4 genotypes, and cardiovascular risk factors.

## What do these findings mean?

- These findings strengthen the contribution of outdoor air pollution on cognitive aging.

- These results highlight the potential benefits of reducing outdoor air pollution levels.

- Key study limitations include measurement error in exposure estimates, potential unmeasured confounders, and limited generalizability.

## Introduction

A growing body of epidemiological evidence supports late-life exposure to ambient air pollutants as an important modifiable risk factor for dementia [1]. These human data converge with neurotoxicological studies that demonstrate increased early markers of neurodegenerative disease (accumulation of amyloid-β and phosphorylation of tau), changes in hippocampal neuronal morphology, and increased cognitive deficits in animals with inhaled exposures to particles [2–8]. Neuroimaging studies in humans have also reported associations between increased fine particulate matter ($PM_{2.5}$; aerodynamic diameter <2.5 μm) and nitrogen dioxide ($NO_2$) and smaller brain volumes in gray matter [9–16], including areas vulnerable to Alzheimer disease (AD) neuropathologies [17,18]. Despite the suggestive evidence for a possible continuum of air pollution neurotoxicity on brain aging processes [19], the reported associations between exposures and cognitive decline have been mixed [20].

Since National Ambient Air Quality Standards were first put into effect 50 years ago, significant reductions in air pollution levels have been seen throughout the US [21]. During the period of 2000 to 2010, annual averages of $PM_{2.5}$ and $NO_2$ decreased by 27% and 35%, respectively [22]. Long-term reduction in the ambient levels of these air pollutants have been linked with increased life expectancy [23], reduced mortality [24], and improved respiratory health (lung function growth, decreased bronchitic symptoms, and lower asthma incidence) [25–27].

Decreasing air pollution levels across the nation provide the ideal environmental context for a quasi-experimental approach [28] to studying the potential benefits of improved air quality (AQ) on maintaining brain health of older people. To the best of our knowledge, no previous studies have explored whether AQ improvement may be associated with cognitive function. We leveraged a nationwide cohort of community-dwelling older women with

individual-level air pollution exposure estimates (1996 to 2012) and annual assessments of late-life cognitive function (2008 to 2018). We hypothesized that improved AQ, as indicated by reductions in $PM_{2.5}$ and $NO_2$, was associated with slower rate of cognitive decline in older women. We further explored whether the putative associations might differ by age, region, education, Apolipoprotein E (ApoE) e4 genotype, and cardiovascular risk factors.

## Methods

### Study design and population

We conducted a longitudinal study on a geographically diverse cohort of community-dwelling older women (*N* = 2,880; aged 74 to 92 years) enrolled in the Women's Health Initiative (WHI) Memory Study (WHIMS)-Epidemiology of Cognitive Health Outcomes (WHIMS-E-CHO) study. WHIMS-ECHO began in 2008 as an extension study of WHIMS [29]—an ancillary study to the Women's Health Initiative hormone therapy (WHI-HT) trials (1993 to 2004). WHIMS participants (*N* = 7,479) were community-dwelling postmenopausal women without dementia (aged ≥65 years) who resided in the 48 contiguous US states and were recruited from more than 40 study sites located in 24 states and Washington, DC. WHIMS-ECHO participants received annual neuropsychological assessments via centralized telephone-administered cognitive interviews conducted by trained and certified staff. The analyses were restricted to women without prevalent dementia at WHIMS-ECHO enrollment and with follow-up visits and complete data on AQ measures and relevant covariates.

Our study did not employ a prospective protocol. Analyses were first planned and performed in April 2020, and before the submission, the completed manuscript was revised by 2 anonymous reviewers assigned by the WHI Publications and Presentations Committee. During the peer review process, we added a partially adjusted model in our main analyses and ad hoc analyses to explore the potential impact of nonlinear cognitive trajectory and nonlinear AQ improvement effects on cognitive trajectory slope. The Institutional Review Board at the University of Southern California reviewed and approved all study protocols. Written informed consent was obtained from all participants as part of WHI-HT, WHIMS, and WHIMS-ECHO studies. This study is reported as per the Strengthening the Reporting of Observational Studies in Epidemiology (STROBE) guideline (S1 Checklist).

### Measures of general cognitive status and episodic memory

General cognitive status was assessed with modified Telephone Interview for Cognitive Status (TICSm), which is a widely used screening tool for cognitive impairment in older adults [30]. The TICSm includes 16 cognitive test items from different cognitive function domains, including recall of a list of words, generating a list of nouns as quickly as possible, naming common items, and counting backwards. The TICSm score (0 to 50) was defined as the total number of correct responses, with test items with multiple parts contributing a corresponding number of points and higher scores indicating better cognitive functioning. Episodic memory, one of the most sensitive cognitive indicators for early detection of AD [31], was assessed by the telephone-based California Verbal Learning Test (CVLT) [32]. Participants were read a 16-item list of words from 4 semantically related categories and were instructed to immediately repeat back as many words as could be remembered. We used the modified version of the CVLT with 3 immediate free recall trials. The CVLT score was defined as the total number of correct responses across 3 learning trials (ranged from 0 to 48), with higher scores representing better performance. In the present study, we used all longitudinal data collected from telephone-based assessments until June 2018 [29].

## Estimation of air pollution exposure

Participants' residential addresses were prospectively collected at each WHI assessment since its inception in 1993, updated at least biannually, and then geocoded using standardized procedures [33]. The exact date of address change was used in analyses when available; otherwise, the date when the change in residence was ascertained was used. We used validated regionalized national universal kriging models with partial least squares regression of geographic covariates and US Environmental Protection Agency (EPA) monitoring data to estimate annual mean concentrations of $PM_{2.5}$ (in μg/m$^3$) and $NO_2$ (in parts per billion [ppb]; a proxy of traffic-related air pollutants) at each of these residential addresses. Over 300 geographic covariates covering land use, vegetative index, proximity to features, etc., were used in the historical models for pre-1999 $PM_{2.5}$ estimation or in the national models for post-1999 $PM_{2.5}$ estimation (average cross-validation $R^2 = 0.88$) [34,35]. Over 400 geographic covariates representing proximity and buffer measures as well as satellite-derived $NO_2$ data were used to estimate $NO_2$ (average cross-validation $R^2 = 0.85$) [36]. These annual estimates were then aggregated to the 3-year average at the WHIMS-ECHO enrollment (referred to as recent exposure) and the corresponding 3-year average 10 years earlier (referred to as remote exposure), accounting for residential mobility within each 3-year time window. The individual-level measure of long-term AQ improvement over the 10-year period was defined as the reduction from remote to recent exposures (Fig 1). We focused on AQ improvement over the 10 years prior to recruitment in order to avoid methodological concerns on the temporality between the defined period of AQ improvement and the concurrently observed slower cognitive decline during the follow-up.

## Covariate data

Participants completed structured questionnaires at WHI inception to provide information on demographics (geographic region where participants resided, age, and race/ethnicity [non-Hispanic Black, non-Hispanic White, or others including Hispanic/Latino and missing]), socioeconomic factors (education, family income, and employment status), and lifestyle factors (smoking status, alcohol intake, and physical activity). Clinical characteristics included body mass index (BMI; calculated from measured height and weight), self-reported use of any postmenopausal hormone treatment, depressive symptoms (assessed using the Center for Epidemiologic Studies Depression Scale short form), and self-reported histories of cardiovascular diseases (CVDs; e.g., heart problems, problems with blood circulation, or blood clots), hypercholesterolemia, hypertension, and diabetes mellitus. Good reliability and validity of both the self-reported medical histories and the physical measures have been documented [37–39]. Lifestyle factors and clinical covariates (BMI; blood pressure; and incident CVD events) were also updated before the WHIMS-ECHO enrollment. Socioeconomic characteristics of residential neighborhoods were characterized using US Census tract-level residential data and estimated at both WHI inception and WHIMS-ECHO enrollment [40]. ApoE e4 genotype data were obtained for a subset of women ($n = 1,611$). Details on covariates are available in S1–S5 Texts.

## Statistical analyses

We used linear mixed effect models to examine whether AQ improvement before WHIMS-ECHO enrollment was associated with average decline rates in the TICSm and CVLT trajectories during the follow-up. Each model included a product term of time and AQ improvement, with years since WHIMS-ECHO enrollment used as the timescale (S6 Text). To account for selective attrition over the WHIMS-ECHO follow-up, models were adjusted for time-varying propensity scores (S7 Text). Potential confounders included demographic variables,

(A) Flow chart of study population

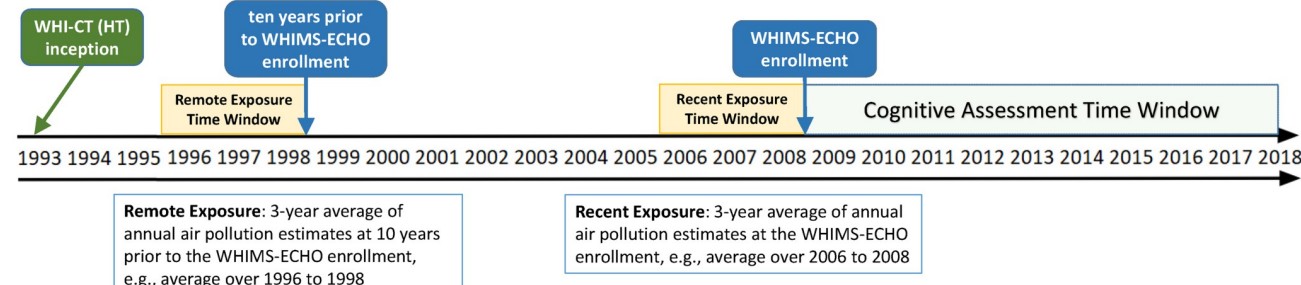

**Fig 1. Flowchart of study population and demonstration of study timeline.** (A) Flowchart of study population. (B) Demonstration of study timeline.[a]
[a]The exposure time windows may vary slightly depending on each individual's WHIMS-ECHO enrollment time. CVLT, California Verbal Learning Test; TICSm, modified Telephone Interview for Cognitive Status; WHI-CT (HT), Women's Health Initiative-Clinical Trial (Hormone Therapy); WHIMS-ECHO, Women's Health Initiative Memory Study-Epidemiology of Cognitive Health Outcomes.

individual- and neighborhood-level socioeconomic characteristics, lifestyle factors, and clinical characteristics at the WHI inception. A random effect for the WHI clinic sites ($n = 39$, S1 Fig) and an indicator of WHIMS-ECHO enrollment year were also included in all models to control for spatial confounding and temporal trends, respectively.

Sensitivity analyses were conducted to evaluate the robustness of our findings. First, to address possible residual confounding due to temporal misspecification of potential

confounders (lifestyle factors, neighborhood socioeconomic characteristics, and clinical attributes), we re-fitted the linear mixed effects models with adjustment of either the measures updated before WHIMS-ECHO enrollment or the changes in these relevant covariates since WHI inception. Second, we applied multiple imputation to address missing data on AQ measures or covariates (S8 Text). Third, to examine whether our findings could be explained by regression to the mean in AQ measures, we further adjusted for recent or remote exposures. Fourth, to explore whether our findings could be explained by cerebrovascular risk, we excluded women with prevalent or incident stroke during the WHIMS-ECHO follow-up. Fifth, we excluded incident dementia cases to further explore whether any observed associations with AQ improvement could be explained by underlying dementia risk (S9 Text). Sixth, we conducted additional analyses with interactions between the measures of AQ improvement and a quadratic term of follow-up year in the adjusted models to examine whether AQ improvement was associated with nonlinear changes in cognitive function. We then evaluated whether the associations between AQ improvement and linear trajectory slopes were sensitive to incorporating the nonlinear changes of cognitive function in the analyses with a quadratic term of follow-up year. Finally, we assessed the nonlinear AQ improvement effect on cognitive trajectory slope by examining the interaction between a quadratic term of AQ improvement and follow-up year.

We also explored whether the putative slower cognitive decline associated with improved AQ might differ by age, education, geographical region, ApoE e4 genotypes, and cardiovascular risk factors, using a product term of the AQ improvement indicator, follow-up time, and each potential effect modifier.

All statistical analyses were performed using R version 3.6.2 and SAS 9.4 for Windows. All tests were interpreted at the 0.05 significance level using a 2-sided alternative hypothesis.

## Results

We excluded 346 women with prevalent dementia at WHIMS-ECHO enrollment or without follow-up visits, resulting in a sample of 2,534 women with at least 2 modified TICSm measures of general cognitive status (Fig 1). For the analyses on episodic memory decline, we further excluded 587 women without repeated measures of episodic memory assessed by CVLT. For both sets of analyses, we also excluded women with missing data on AQ measures or relevant covariates. This resulted in a final analytic sample of 2,232 women for the analyses on general cognitive status assessed by TICSm, a subset of which ($n$ = 1,721) was used for the analyses on episodic memory assessed by CVLT (Fig 1).

Compared to the women excluded due to missing AQ measures or relevant covariates (Fig 1), women included in our analyses were younger with higher socioeconomic status and more likely to reside in the Northeast, self-identify as non-Hispanic White, and drink alcohol and have ApoE e4 genotype (Table 1). Compared to those excluded due to not having repeated CVLT measures (Fig 1), women with repeated CVLT measures were more likely to be younger, have higher education and income, and currently drink alcohol, but less likely to report having hypertension and carry the ApoE e4 genotype (S1 Table). Mean AQ improved significantly with reduced ambient levels for both $PM_{2.5}$ (Mean ± SD: 13.3 ± 2.7 to 10.6 ± 2.0 μg/m³; $p < 0.001$) and $NO_2$ (15.7 ± 7.2 to 10.4 ± 4.9 ppb; $p < 0.001$) over the 10 years before WHIMS-ECHO enrollment. Women residing in locations with initially high ambient air pollutants tended to experience greater AQ improvement for both $PM_{2.5}$ (correlation = 0.67; $p < 0.001$) and $NO_2$ (correlation = 0.80; $p < 0.001$) (S2 Table). Overall, non-Hispanic White women experienced less AQ improvement, while women who were older than 80, reported higher income, or resided in the Northeast and West experienced greater reductions in ambient $PM_{2.5}$ and $NO_2$, as compared to their counterparts (Table 2). Women with ApoE e4 genotype

**Table 1. Distribution of population characteristics in the WHIMS-ECHO cohort, stratified by women included versus excluded.**

| Characteristics | Analytic sample for TICSm analyses | | | | Analytic sample for CVLT analyses | | | |
|---|---|---|---|---|---|---|---|---|
| | Study sample (N = 2,534)[a] | Included (N = 2,232) | Excluded (N = 302)[a] | p[b] | Study sample (N = 1,947)[a] | Included (N = 1,721) | Excluded (N = 226)[a] | p[b] |
| **Region** | | | | <0.001 | | | | <0.001 |
| Northeast | 773 (30.5%) | 718 (32.2%) | 55 (18.2%) | | 593 (30.5%) | 551 (32.0%) | 42 (18.6%) | |
| South | 540 (21.3%) | 443 (19.8%) | 97 (32.1%) | | 402 (20.6%) | 329 (19.1%) | 73 (32.3%) | |
| Midwest | 618 (24.4%) | 549 (24.6%) | 69 (22.8%) | | 482 (24.8%) | 428 (24.9%) | 54 (23.9%) | |
| West | 603 (23.8%) | 522 (23.4%) | 81 (26.8%) | | 470 (24.1%) | 413 (24.0%) | 57 (25.2%) | |
| **Age** | | | | 0.04 | | | | 0.02 |
| ≤80 years | 979 (38.6%) | 879 (39.4%) | 100 (33.1%) | | 808 (41.5%) | 731 (42.5%) | 77 (34.1%) | |
| >80 years | 1,555 (61.4%) | 1,353 (60.6%) | 202 (66.9%) | | 1,139 (58.5%) | 990 (57.5%) | 149 (65.9%) | |
| **Ethnicity** | | | | <0.001 | | | | <0.001 |
| Black (not Hispanic) | 160 (6.3%) | 116 (5.2%) | 44 (14.6%) | | 120 (6.2%) | 86 (5.0%) | 34 (15.0%) | |
| White (not Hispanic) | 2,262 (89.3%) | 2,042 (91.5%) | 220 (72.8%) | | 1,742 (89.5%) | 1,576 (91.6%) | 166 (73.5%) | |
| Other or missing | 112 (4.4%) | 74 (3.3%) | 38 (12.6%) | | 85 (4.4%) | 59 (3.4%) | 26 (11.5%) | |
| **Education** | | | | 0.03 | | | | 0.02 |
| ≤High school or GED | 660 (26.1%) | 564 (25.3%) | 96 (32.2%) | | 476 (24.5%) | 406 (23.6%) | 70 (31.6%) | |
| >High school but <4 years of college | 975 (38.5%) | 864 (38.7%) | 111 (37.2%) | | 733 (37.7%) | 651 (37.8%) | 82 (36.9%) | |
| ≥4 years of college | 895 (35.4%) | 804 (36.0%) | 91 (30.5%) | | 734 (37.8%) | 664 (38.6%) | 70 (31.5%) | |
| **Employment** | | | | 0.69 | | | | 0.72 |
| Currently working | 388 (15.4%) | 348 (15.6%) | 40 (13.7%) | | 310 (16.0%) | 279 (16.2%) | 31 (14.1%) | |
| Not working | 238 (9.4%) | 210 (9.4%) | 28 (9.6%) | | 188 (9.7%) | 166 (9.6%) | 22 (10.0%) | |
| Retired | 1,899 (75.2%) | 1,674 (75.0%) | 225 (76.8%) | | 1,443 (74.3%) | 1,276 (74.1%) | 167 (75.9%) | |
| **Income (US$)** | | | | 0.03 | | | | 0.02 |
| <9,999 | 94 (3.7%) | 74 (3.3%) | 20 (6.6%) | | 67 (3.4%) | 52 (3.0%) | 15 (6.6%) | |
| 10,000 to 34,999 | 1,140 (45.0%) | 1,000 (44.8%) | 140 (46.4%) | | 849 (43.6%) | 742 (43.1%) | 107 (47.3%) | |
| 35,000 to 74,999 | 927 (36.6%) | 824 (36.9%) | 103 (34.1%) | | 741 (38.1%) | 663 (38.5%) | 78 (34.5%) | |
| 75,000 or more | 242 (9.6%) | 220 (9.9%) | 22 (7.3%) | | 204 (10.5%) | 186 (10.8%) | 18 (8.0%) | |
| Do not know | 131 (5.2%) | 114 (5.1%) | 17 (5.6%) | | 86 (4.4%) | 78 (4.5%) | 8 (3.5%) | |
| **Smoking status** | | | | 0.38 | | | | 0.23 |
| Never smoked | 1,396 (55.6%) | 1,239 (55.5%) | 157 (55.9%) | | 1,074 (55.7%) | 954 (55.4%) | 120 (57.7%) | |
| Past smoker | 996 (39.6%) | 890 (39.9%) | 106 (37.7%) | | 763 (39.6%) | 689 (40.0%) | 74 (35.6%) | |
| Current smoker | 121 (4.8%) | 103 (4.6%) | 18 (6.4%) | | 92 (4.8%) | 78 (4.5%) | 14 (6.7%) | |
| **Alcohol use** | | | | 0.004 | | | | 0.002 |
| Nondrinker | 307 (12.2%) | 261 (11.7%) | 46 (16.0%) | | 227 (11.7%) | 191 (11.1%) | 36 (16.7%) | |
| Past drinker | 436 (17.3%) | 372 (16.7%) | 64 (22.3%) | | 313 (16.2%) | 266 (15.5%) | 47 (21.8%) | |
| <1 drink per day | 1,466 (58.2%) | 1,315 (58.9%) | 151 (52.6%) | | 1,150 (59.4%) | 1,036 (60.2%) | 114 (52.8%) | |
| ≥1 drink per day | 310 (12.3%) | 284 (12.7%) | 26 (9.1%) | | 247 (12.8%) | 228 (13.2%) | 19 (8.8%) | |
| **Moderate or strenuous physical activities ≥20 minutes** | | | | 0.25 | | | | 0.17 |
| No activity | 1,387 (54.8%) | 1,207 (54.1%) | 180 (60.0%) | | 1,053 (54.1%) | 917 (53.3%) | 136 (60.7%) | |
| Some activity | 138 (5.5%) | 124 (5.6%) | 14 (4.7%) | | 102 (5.2%) | 91 (5.3%) | 11 (4.9%) | |
| 2 to 4 episodes/week | 534 (21.1%) | 481 (21.6%) | 53 (17.7%) | | 413 (21.2%) | 376 (21.8%) | 37 (16.5%) | |
| >4 episodes/week | 473 (18.7%) | 420 (18.8%) | 53 (17.7%) | | 377 (19.4%) | 337 (19.6%) | 40 (17.9%) | |
| **BMI (kg/m$^2$)** | | | | 0.36 | | | | 0.67 |
| <25 | 701 (27.8%) | 619 (27.7%) | 82 (28.3%) | | 538 (27.7%) | 476 (27.7%) | 62 (28.4%) | |
| 25 to 29 | 931 (36.9%) | 815 (36.5%) | 116 (40.0%) | | 718 (37.0%) | 633 (36.8%) | 85 (39.0%) | |
| ≥30 | 890 (35.3%) | 798 (35.8%) | 92 (31.7%) | | 683 (35.2%) | 612 (35.6%) | 71 (32.6%) | |

*(Continued)*

**Table 1.** (Continued)

| Characteristics | Analytic sample for TICSm analyses | | | | Analytic sample for CVLT analyses | | | |
|---|---|---|---|---|---|---|---|---|
| | Study sample (N = 2,534)[a] | Included (N = 2,232) | Excluded (N = 302)[a] | p[b] | Study sample (N = 1,947)[a] | Included (N = 1,721) | Excluded (N = 226)[a] | p[b] |
| **Hypertension** | | | | 0.96 | | | | 0.56 |
| No | 1,646 (65.5%) | 1,461 (65.5%) | 185 (65.6%) | | 1,289 (66.7%) | 1,152 (66.9%) | 137 (64.9%) | |
| Yes | 868 (34.5%) | 771 (34.5%) | 97 (34.4%) | | 643 (33.3%) | 569 (33.1%) | 74 (35.1%) | |
| **Hypercholesterolemia** | | | | 0.06 | | | | 0.10 |
| No | 2,063 (82.6%) | 1,855 (83.1%) | 208 (78.5%) | | 1,595 (83.1%) | 1,438 (83.6%) | 157 (78.9%) | |
| Yes | 434 (17.4%) | 377 (16.9%) | 57 (21.5%) | | 325 (16.9%) | 283 (16.4%) | 42 (21.1%) | |
| **Diabetes** | | | | 0.41 | | | | 0.12 |
| No | 2,433 (96.1%) | 2,143 (96.0%) | 290 (97.0%) | | 1,876 (96.4%) | 1,655 (96.2%) | 221 (98.2%) | |
| Yes | 98 (3.9%) | 89 (4.0%) | 9 (3.0%) | | 70 (3.6%) | 66 (3.8%) | 4 (1.8%) | |
| **CVD history** | | | | 0.25 | | | | 0.30 |
| No | 2,143 (85.7%) | 1,907 (85.4%) | 236 (88.1%) | | 1,652 (86.0%) | 1,476 (85.8%) | 176 (88.4%) | |
| Yes | 357 (14.3%) | 325 (14.6%) | 32 (11.9%) | | 268 (14.0%) | 245 (14.2%) | 23 (11.6%) | |
| **Any prior postmenopausal hormone treatment** | | | | 0.31 | | | | 0.51 |
| No | 1,373 (54.2%) | 1,218 (54.6%) | 155 (51.5%) | | 1,052 (54.1%) | 935 (54.3%) | 117 (52.0%) | |
| Yes | 1,160 (45.8%) | 1,014 (45.4%) | 146 (48.5%) | | 894 (45.9%) | 786 (45.7%) | 108 (48.0%) | |
| **WHI hormone therapy assignment** | | | | 0.75 | | | | 0.94 |
| CEE alone placebo | 459 (18.1%) | 400 (17.9%) | 59 (19.5%) | | 346 (17.8%) | 306 (17.8%) | 40 (17.7%) | |
| CEE alone | 461 (18.2%) | 403 (18.1%) | 58 (19.2%) | | 345 (17.7%) | 302 (17.5%) | 43 (19.0%) | |
| CEE+MPA placebo | 832 (32.8%) | 733 (32.8%) | 99 (32.8%) | | 654 (33.6%) | 578 (33.6%) | 76 (33.6%) | |
| CEE+MPA | 782 (30.9%) | 696 (31.2%) | 86 (28.5%) | | 602 (30.9%) | 535 (31.1%) | 67 (29.6%) | |
| **ApoE[c]** | | | | 0.06 | | | | 0.04 |
| e2/2+e2/3+e3/3 | 1,382 (77.5%) | 1,239 (76.9%) | 143 (83.1%) | | 1,097 (79.3%) | 984 (78.5%) | 113 (86.3%) | |
| e2/4+e3/4+e4/4 | 401 (22.5%) | 372 (23.1%) | 29 (16.9%) | | 287 (20.7%) | 269 (21.5%) | 18 (13.7%) | |

[a]Numbers in the samples may not add up to total due to missing data.

[b]p-Values were calculated using chi-squared tests.

[c]Numbers in the samples with ApoE genotyping did not add up to the total due to missing data.

ApoE, Apolipoprotein E; BMI, body mass index; CEE, conjugated equine estrogens; CVD, cardiovascular disease; CVLT, California Verbal Learning Test; GED, general educational development; MPA, medroxyprogesterone acetate; $NO_2$, nitrogen dioxide; $PM_{2.5}$, fine particulate matter; SD, standard deviation; TICSm, modified Telephone Interview for Cognitive Status; WHI, Women's Health Initiative; WHIMS-ECHO, Women's Health Initiative Memory Study-Epidemiology of Cognitive Health Outcomes.

experienced less reduction in ambient $PM_{2.5}$, while nondrinkers experienced less reduction in ambient $NO_2$, as compared to their counterparts (Table 2).

During a median 6.2 (interquartile range [IQR] = 5) years of follow-up, women had a median 7 (IQR = 5) interviews for TICSm tests and 6 (IQR = 3) interviews for CVLT tests. The means of the cognitive test scores did not differ much over the visits (S3 Table), while significant cognitive declines (TICSm slope = −0.42/year; 95% CI: −0.44, −0.40; CVLT slope = −0.59/year; 95% CI: −0.64, −0.54) were observed (Fig 2). Non-Hispanic Black women or women older than 80 had lower mean cognitive scores at baseline for both TICSm and CVLT (S4 Table). Women residing in the Midwest and who had higher education or income, were currently employed, currently drinking alcohol, did not have hypertension, and did not carry the ApoE e4 genotype had higher mean cognitive scores on both TICSm and CVLT at baseline, compared to their counterparts (S4 Table). The distributions of cognitive scores at the last visit

 

**Table 2. Distribution of AQ measures by population characteristics in the WHIMS-ECHO cohort, 1998 to 2018.**

| | | Analytic sample for TICSm analyses | | | | | Analytic sample for CVLT analyses | | | |
| | | AQ improvement in PM$_{2.5}$ (μg/m$^3$)[a] | | AQ improvement in NO$_2$ (ppb)[a] | | | AQ improvement in PM$_{2.5}$ (μg/m$^3$)[a] | | AQ improvement in NO$_2$ (ppb)[a] | |
| Population characteristics | N | Mean ± SD | $p$[b] | Mean ± SD | $p$[b] | N | Mean ± SD | $p$[b] | Mean ± SD | $p$[b] |
|---|---|---|---|---|---|---|---|---|---|---|
| **Overall** | 2,232 | 2.73 ± 1.63 | | 5.27 ± 3.46 | | 1,721 | 2.78 ± 1.68 | | 5.34 ± 3.49 | |
| **Region** | | | <0.001 | | <0.001 | | | <0.001 | | <0.001 |
| Northeast | 718 | 3.01 ± 0.95 | | 5.72 ± 3.26 | | 551 | 3.02 ± 0.92 | | 5.73 ± 3.23 | |
| South | 443 | 2.50 ± 1.28 | | 4.92 ± 3.08 | | 329 | 2.54 ± 1.26 | | 5.02 ± 3.08 | |
| Midwest | 549 | 2.15 ± 1.22 | | 4.36 ± 2.32 | | 428 | 2.12 ± 1.21 | | 4.36 ± 2.25 | |
| West | 522 | 3.18 ± 2.55 | | 5.89 ± 4.60 | | 413 | 3.35 ± 2.63 | | 6.08 ± 4.73 | |
| **Age** | | | 0.009 | | 0.004 | | | 0.002 | | 0.002 |
| ≤80 years | 879 | 2.62 ± 1.50 | | 5.00 ± 3.23 | | 731 | 2.64 ± 1.54 | | 5.03 ± 3.32 | |
| >80 years | 1,353 | 2.81 ± 1.71 | | 5.44 ± 3.58 | | 990 | 2.89 ± 1.76 | | 5.56 ± 3.60 | |
| **Ethnicity** | | | <0.001 | | <0.001 | | | <0.001 | | <0.001 |
| Black (not Hispanic) | 116 | 3.28 ± 1.39 | | 6.84 ± 2.67 | | 86 | 3.49 ± 1.31 | | 6.91 ± 2.64 | |
| White (not Hispanic) | 2,042 | 2.68 ± 1.63 | | 5.13 ± 3.47 | | 1,576 | 2.72 ± 1.67 | | 5.21 ± 3.51 | |
| Other | 74 | 3.42 ± 1.84 | | 6.42 ± 3.30 | | 59 | 3.51 ± 1.94 | | 6.46 ± 3.39 | |
| **Education** | | | 0.04 | | 0.09 | | | 0.06 | | 0.48 |
| ≤High school or GED | 564 | 2.64 ± 1.49 | | 5.08 ± 3.22 | | 406 | 2.65 ± 1.51 | | 5.17 ± 3.32 | |
| >High school but <4 years of college | 864 | 2.69 ± 1.77 | | 5.19 ± 3.62 | | 651 | 2.75 ± 1.82 | | 5.34 ± 3.74 | |
| ≥4 years of college | 804 | 2.85 ± 1.57 | | 5.47 ± 3.43 | | 664 | 2.89 ± 1.63 | | 5.44 ± 3.35 | |
| **Employment** | | | 0.81 | | 0.33 | | | 0.84 | | 0.49 |
| Currently working | 348 | 2.78 ± 1.63 | | 5.51 ± 3.52 | | 279 | 2.81 ± 1.68 | | 5.57 ± 3.62 | |
| Not working | 210 | 2.76 ± 1.69 | | 5.13 ± 3.64 | | 166 | 2.84 ± 1.76 | | 5.32 ± 3.77 | |
| Retired | 1,674 | 2.72 ± 1.63 | | 5.23 ± 3.42 | | 1,276 | 2.77 ± 1.67 | | 5.29 ± 3.43 | |
| **Income (US$)** | | | 0.01 | | 0.04 | | | 0.04 | | 0.03 |
| <9,999 | 74 | 2.73 ± 1.98 | | 5.02 ± 3.93 | | 52 | 2.72 ± 2.10 | | 5.00 ± 3.97 | |
| 10,000 to 34,999 | 1,000 | 2.64 ± 1.66 | | 5.18 ± 3.54 | | 742 | 2.73 ± 1.70 | | 5.30 ± 3.50 | |
| 35,000 to 74,999 | 824 | 2.78 ± 1.55 | | 5.22 ± 3.24 | | 663 | 2.79 ± 1.61 | | 5.21 ± 3.37 | |
| 75,000 or more | 220 | 3.07 ± 1.73 | | 5.95 ± 3.88 | | 186 | 3.10 ± 1.76 | | 6.10 ± 3.91 | |
| Do not know | 114 | 2.55 ± 1.43 | | 5.11 ± 2.88 | | 78 | 2.48 ± 1.36 | | 5.17 ± 2.80 | |
| **Smoking status** | | | 0.98 | | 0.66 | | | 0.80 | | 0.67 |
| Never smoked | 1,239 | 2.74 ± 1.67 | | 5.27 ± 3.38 | | 954 | 2.80 ± 1.71 | | 5.38 ± 3.44 | |
| Past smoker | 890 | 2.73 ± 1.59 | | 5.23 ± 3.56 | | 689 | 2.75 ± 1.65 | | 5.25 ± 3.57 | |
| Current smoker | 103 | 2.76 ± 1.54 | | 5.55 ± 3.52 | | 78 | 2.83 ± 1.56 | | 5.52 ± 3.42 | |
| **Alcohol use** | | | 0.48 | | <0.001 | | | 0.54 | | 0.006 |
| Nondrinker | 261 | 2.59 ± 1.72 | | 4.60 ± 3.18 | | 191 | 2.62 ± 1.78 | | 4.63 ± 3.30 | |
| Past drinker | 372 | 2.75 ± 1.67 | | 5.37 ± 3.78 | | 266 | 2.85 ± 1.71 | | 5.52 ± 3.92 | |
| <1 drink per day | 1,315 | 2.76 ± 1.60 | | 5.46 ± 3.41 | | 1,036 | 2.79 ± 1.65 | | 5.49 ± 3.43 | |
| ≥1 drink per day | 284 | 2.71 ± 1.67 | | 4.85 ± 3.36 | | 228 | 2.79 ± 1.67 | | 5.02 ± 3.35 | |
| **Moderate or strenuous physical activities ≥20 minutes** | | | 0.82 | | 0.25 | | | 0.74 | | 0.21 |
| No activity | 1,207 | 2.71 ± 1.63 | | 5.28 ± 3.41 | | 917 | 2.78 ± 1.68 | | 5.42 ± 3.51 | |
| Some activity | 124 | 2.76 ± 1.24 | | 5.56 ± 3.18 | | 91 | 2.75 ± 1.23 | | 5.58 ± 3.16 | |
| 2 to 4 episodes/week | 481 | 2.79 ± 1.68 | | 5.38 ± 3.54 | | 376 | 2.85 ± 1.74 | | 5.40 ± 3.51 | |
| >4 episodes/week | 420 | 2.72 ± 1.69 | | 4.99 ± 3.56 | | 337 | 2.72 ± 1.70 | | 4.98 ± 3.51 | |
| **BMI (kg/m$^2$)** | | | 0.43 | | 0.48 | | | 0.91 | | 0.09 |
| <25 | 619 | 2.80 ± 1.64 | | 5.19 ± 3.37 | | 476 | 2.81 ± 1.70 | | 5.10 ± 3.34 | |
| 25 to 29 | 815 | 2.74 ± 1.64 | | 5.21 ± 3.25 | | 633 | 2.76 ± 1.68 | | 5.30 ± 3.29 | |

*(Continued)*

 

**Table 2.** (Continued)

| | | Analytic sample for TICSm analyses | | | | Analytic sample for CVLT analyses | | | |
|---|---|---|---|---|---|---|---|---|---|
| | | AQ improvement in PM$_{2.5}$ (μg/m³)[a] | | AQ improvement in NO$_2$ (ppb)[a] | | | AQ improvement in PM$_{2.5}$ (μg/m³)[a] | | AQ improvement in NO$_2$ (ppb)[a] | |
| **Population characteristics** | N | Mean ± SD | $p$[b] | Mean ± SD | $p$[b] | N | Mean ± SD | $p$[b] | Mean ± SD | $p$[b] |
| ≥30 | 798 | 2.68 ± 1.62 | | 5.38 ± 3.72 | | 612 | 2.78 ± 1.66 | | 5.56 ± 3.79 | |
| **Hypertension** | | | 0.82 | | 0.52 | | | 0.63 | | 0.31 |
| No | 1,461 | 2.73 ± 1.65 | | 5.23 ± 3.52 | | 1,152 | 2.77 ± 1.71 | | 5.28 ± 3.54 | |
| Yes | 771 | 2.75 ± 1.60 | | 5.33 ± 3.33 | | 569 | 2.81 ± 1.61 | | 5.46 ± 3.38 | |
| **Hypercholesterolemia** | | | 0.12 | | 0.97 | | | 0.44 | | 0.97 |
| No | 1,855 | 2.71 ± 1.65 | | 5.26 ± 3.53 | | 1,438 | 2.77 ± 1.70 | | 5.34 ± 3.57 | |
| Yes | 377 | 2.85 ± 1.56 | | 5.27 ± 3.04 | | 283 | 2.85 ± 1.56 | | 5.34 ± 3.09 | |
| **Diabetes** | | | 0.27 | | 0.42 | | | 0.07 | | 0.11 |
| No | 2,143 | 2.73 ± 1.64 | | 5.25 ± 3.47 | | 1,655 | 2.77 ± 1.68 | | 5.31 ± 3.50 | |
| Yes | 89 | 2.92 ± 1.50 | | 5.56 ± 3.16 | | 66 | 3.15 ± 1.48 | | 6.02 ± 3.28 | |
| **CVD history** | | | 0.95 | | 0.94 | | | 0.95 | | 0.83 |
| No | 1,907 | 2.74 ± 1.65 | | 5.27 ± 3.52 | | 1,476 | 2.78 ± 1.70 | | 5.34 ± 3.58 | |
| Yes | 325 | 2.73 ± 1.53 | | 5.25 ± 3.04 | | 245 | 2.78 ± 1.52 | | 5.29 ± 2.94 | |
| **Any prior postmenopausal hormone treatment** | | | 0.84 | | 0.22 | | | 0.90 | | 0.29 |
| No | 1,218 | 2.73 ± 1.44 | | 5.35 ± 3.31 | | 935 | 2.79 ± 1.47 | | 5.42 ± 3.29 | |
| Yes | 1,014 | 2.74 ± 1.84 | | 5.17 ± 3.62 | | 786 | 2.78 ± 1.90 | | 5.24 ± 3.72 | |
| **WHI hormone therapy assignment** | | | 0.13 | | 0.02 | | | 0.03 | | 0.09 |
| CEE alone placebo | 400 | 2.81 ± 1.74 | | 5.24 ± 3.33 | | 306 | 2.89 ± 1.84 | | 5.35 ± 3.43 | |
| CEE alone | 403 | 2.57 ± 1.65 | | 4.80 ± 3.30 | | 302 | 2.53 ± 1.69 | | 4.88 ± 3.37 | |
| CEE+MPA placebo | 733 | 2.77 ± 1.55 | | 5.45 ± 3.53 | | 578 | 2.84 ± 1.58 | | 5.47 ± 3.46 | |
| CEE+MPA | 696 | 2.75 ± 1.64 | | 5.35 ± 3.52 | | 535 | 2.79 ± 1.66 | | 5.44 ± 3.61 | |
| **ApoE** | | | 0.01 | | 0.92 | | | 0.047 | | 0.87 |
| e2/2+e2/3+e3/3 | 1,239 | 2.81 ± 1.55 | | 5.33 ± 3.46 | | 984 | 2.82 ± 1.58 | | 5.39 ± 3.45 | |
| e2/4+e3/4+e4/4 | 372 | 2.56 ± 1.77 | | 5.31 ± 3.63 | | 269 | 2.59 ± 1.88 | | 5.35 ± 3.73 | |

[a]Recent exposures were the 3-year average exposures estimated at the WHIMS-ECHO enrollment. Remote exposures were the 3-year average exposures estimated 10 years before the WHIMS-ECHO enrollment. AQ improvement was defined as reduction from the remote to recent exposures over the 10-year period.

[b]$p$-Values were calculated using ANOVA F-tests for mean exposures.

ApoE, Apolipoprotein E; AQ, air quality; BMI, body mass index; CEE, conjugated equine estrogens; CVD, cardiovascular disease; CVLT, California Verbal Learning Test; GED, general educational development; MPA, medroxyprogesterone acetate; NO$_2$, nitrogen dioxide; PM$_{2.5}$, fine particulate matter; ppb, parts per billion; SD, standard deviation; TICSm, modified Telephone Interview for Cognitive Status; WHI, Women's Health Initiative; WHIMS-ECHO, Women's Health Initiative Memory Study-Epidemiology of Cognitive Health Outcomes.

were similar across these population characteristics, except that some differences were no longer significant (S4 Table).

Residing in locations with greater AQ improvement was associated with slower rates of decline in both general cognitive status and episodic memory (Table 3). Based on fully adjusted models (Table 3, Model III), each IQR increment of improved AQ in PM$_{2.5}$ and NO$_2$ (IQR$_{PM2.5}$ = 1.79 μg/m³ for both analytic samples; IQR$_{NO2}$ = 3.92 ppb for TICSm and 3.97 ppb for CVLT analyses) was associated with diminished cognitive declines over time by 0.026 to 0.034/year in TICSm (PM$_{2.5}$: ($\beta$ = 0.026/year, 95% CI: 0.001, 0.05; NO$_2$: $\beta$ = 0.034/year; 95% CI: 0.01, 0.06) and by 0.060 to 0.070/year in CVLT (PM$_{2.5}$: $\beta$ = 0.070/year, 95% CI: 0.02, 0.12; NO$_2$: $\beta$ = 0.060/year; 95% CI: 0.005, 0.12). These putative benefits suggested by the respective associations with TICSm and CVLT were equivalent to slower cognitive declines in women who were 0.9 to 1.2 years and 1.4 to 1.6 years younger at WHIMS-ECHO enrollment.

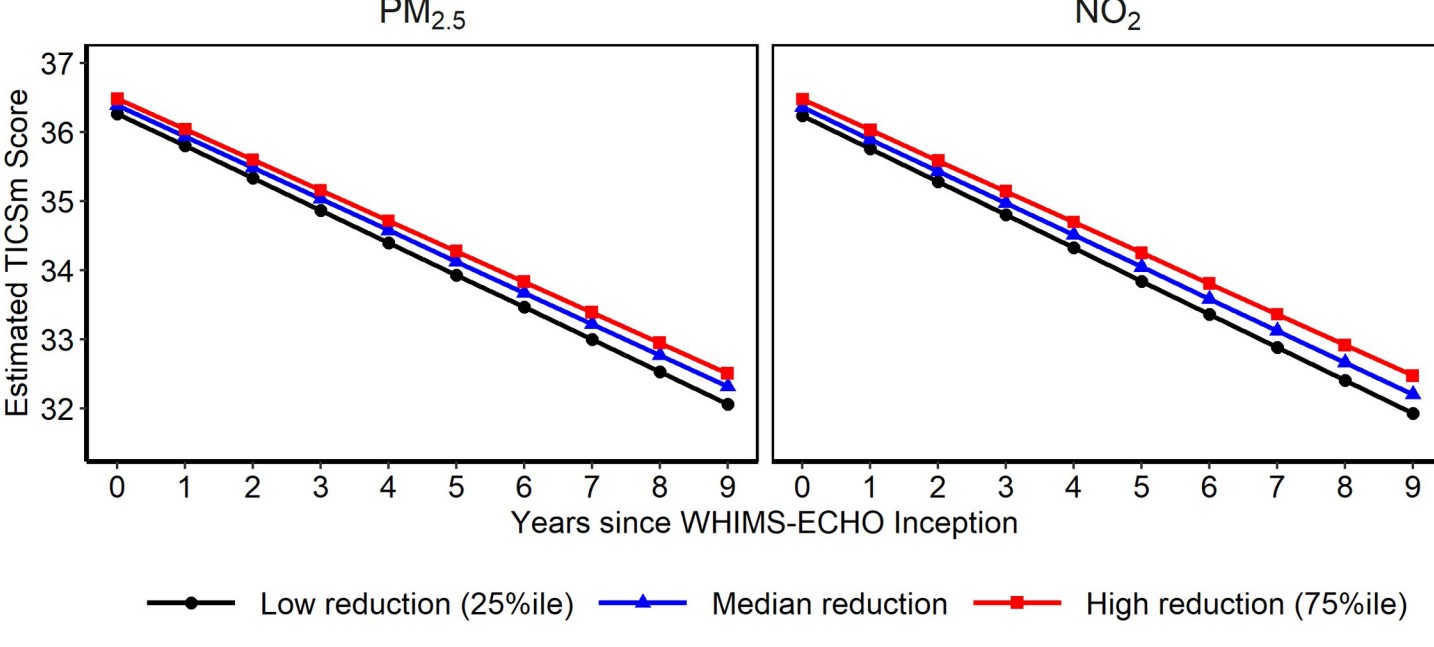

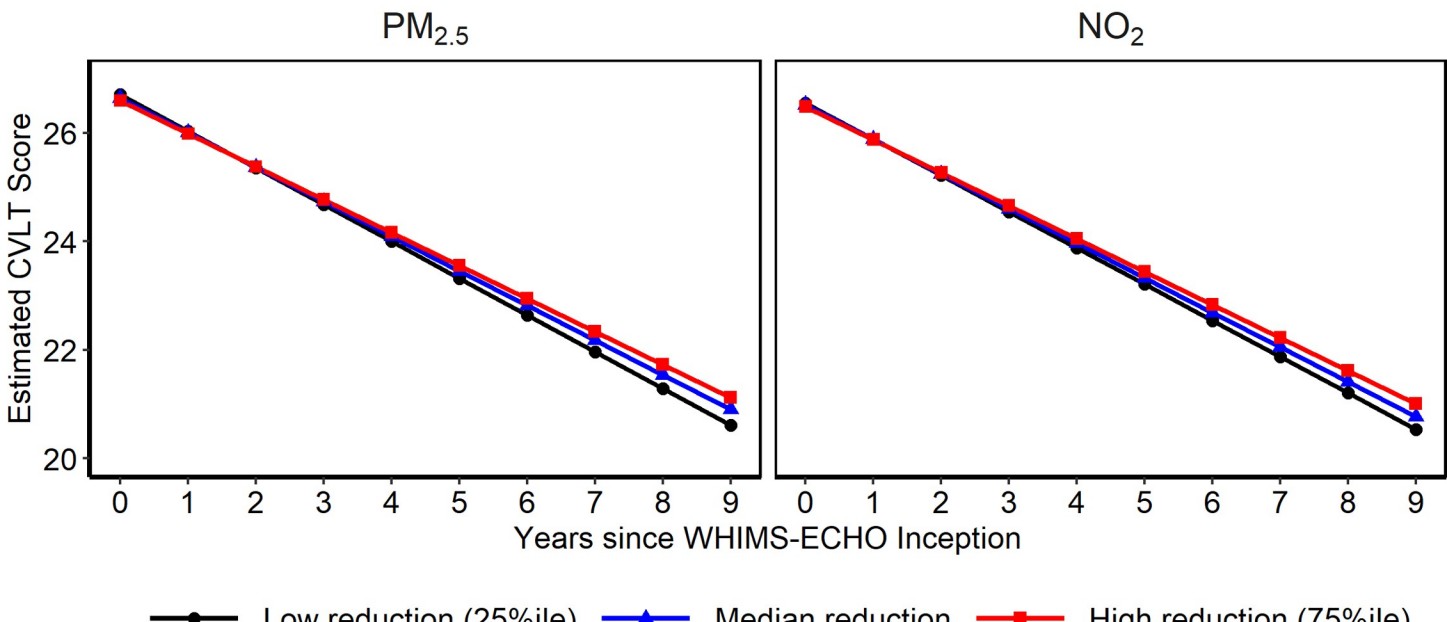

**Fig 2. Estimated cognitive trajectory over time with different levels of AQ improvement in WHIMS-ECHO cohort. (A)** Associations on general cognitive ability decline (N = 2,232). **(B)** Associations on episodic memory decline (N = 1,721). Estimated TICSm score (panel A) or CVLT score (panel B) change over time for low (25th percentile), median, and high (75th percentile) level of AQ improvement in $PM_{2.5}$ or $NO_2$ in the WHIMS-ECHO cohort. The estimated TICSm scores or CVLT scores were calculated using parameter estimates derived from Model III of Table 3, which were adjusted for spatial random effect, WHIMS-ECHO enrollment year, age, follow-up year, age interaction with follow-up year, time-varying propensity scores, demographic variables (geographic region and race/ethnicity), socioeconomic factors (education, income, and employment status) and neighborhood characteristics, lifestyle factors (smoking, drinking, and physical activities), prior hormone use, hormone therapy assignment, cardiovascular risk factors (hypertension, diabetes, and hypercholesterolemia), depression, BMI, and CVD histories. AQ, air quality; BMI, body mass index; CVD, cardiovascular disease; CVLT, California Verbal Learning Test; $NO_2$, nitrogen dioxide; $PM_{2.5}$, fine particulate matter; TICSm, modified Telephone Interview for Cognitive Status; WHIMS-ECHO, Women's Health Initiative Memory Study-Epidemiology of Cognitive Health Outcomes.

**Table 3. Summary of the associations between AQ improvement and cognitive declines in the WHIMS-ECHO cohort.**

**(A) Associations with declines in general cognitive ability (N = 2,232)**

| Models | AQ improvement in $PM_{2.5}$[a] | | | AQ improvement in $NO_2$[a] | | |
|---|---|---|---|---|---|---|
| | β[b] | 95% CI | $p$[c] | β[b] | 95% CI | $p$[c] |
| Model I[d] | 0.026 | 0.001, 0.05 | 0.04 | 0.034 | 0.01, 0.06 | 0.006 |
| Model II[e] | 0.026 | 0.002, 0.05 | 0.04 | 0.034 | 0.01, 0.06 | 0.005 |
| Model III[f] | 0.026 | 0.001, 0.05 | 0.04 | 0.034 | 0.01, 0.06 | 0.005 |

**(B) Associations with declines in episodic memory (N = 1,721)**

| Models | AQ improvement in $PM_{2.5}$[a] | | | AQ improvement in $NO_2$[a] | | |
|---|---|---|---|---|---|---|
| | β[b] | 95% CI | $p$[c] | β[b] | 95% CI | $p$[c] |
| Model I[d] | 0.072 | 0.02, 0.13 | 0.01 | 0.059 | 0.004, 0.11 | 0.03 |
| Model II[e] | 0.071 | 0.02, 0.13 | 0.01 | 0.060 | 0.006, 0.12 | 0.03 |
| Model III[f] | 0.070 | 0.02, 0.12 | 0.01 | 0.060 | 0.005, 0.12 | 0.03 |

[a]Recent exposures were the 3-year average exposures estimated at the WHIMS-ECHO enrollment. Remote exposures were the 3-year average exposures estimated 10 years before the WHIMS-ECHO enrollment. AQ improvement was defined as reduction from the remote to recent exposures over the 10-year period.

[b]β (95% CI) = regression coefficient (95% CI) estimating the increase in TICSm score or CVLT score per year for each IQR increase of AQ improvement ($IQR_{PM2.5}$ = 1.79 μg/m$^3$ for both analytic samples; $IQR_{NO2}$ = 3.92 ppb for TICSm analytic sample and 3.97 ppb for CVLT analytic sample). Positive coefficients represent slower decline associated with greater AQ improvement.

[c]$p$-Values were calculated using Wald $t$ tests.

[d]Model I: adjusted for spatial random effect, WHIMS-ECHO enrollment year, age, follow-up year, age interaction with follow-up year, and time-varying propensity scores.

[e]Model II: adjusted for those in Model I, as well as demographic variables (geographic region and race/ethnicity), socioeconomic factors (education, income, and employment status) and neighborhood socioeconomic characteristics, and lifestyle factors (smoking, drinking, and physical activities).

[f]Model III: adjusted for those in Model II, as well as prior hormone use, hormone therapy assignment, cardiovascular risk factors (hypertension, diabetes, and hypercholesterolemia), depression, BMI, and CVD histories.

AQ, air quality; BMI, body mass index; CVD, cardiovascular disease; CVLT, California Verbal Learning Test; IQR, interquartile range; $NO_2$, nitrogen dioxide; $PM_{2.5}$, fine particulate matter; TICSm, modified Telephone Interview for Cognitive Status; WHIMS-ECHO, Women's Health Initiative Memory Study-Epidemiology of Cognitive Health Outcomes.

The associations between greater AQ improvement and slower cognitive declines remained robust in sensitivity analyses adjusting for potential confounders updated prior to the WHIMS-ECHO enrollment or temporal changes in relevant covariates from WHI inception (S5 Table). Using multiple imputation to include women with missing air pollution and covariate data, the associations were slightly attenuated, but remained statistically significant (S6 Table). The associations between greater AQ improvement and slower cognitive declines were strengthened in models further adjusting for recent or remote exposures (S7 Table). The results were largely unchanged after excluding prevalent or incident stroke cases (S8 Table). Excluding incident dementia cases (n = 398) resulted in a 41% to 81% reduction in the strength of the associations with slower declines in general cognitive status, which were no longer significant. For episodic memory decline, there was a 17% to 22% reduction in effect estimates, which remained statistically significant (S8 Table).

We did not find significant associations between AQ improvement and quadratic change in cognitive function (all $p$-values > 0.10, S9 Table). With the quadratic term of follow-up year included in models, the associations between AQ improvement and TICSm trajectory slope were similar to the estimates in the main analyses, while the estimated associations with CVLT trajectory slopes were slightly attenuated for both $PM_{2.5}$ and $NO_2$ improvement (S10 Table). Except for some evidence supporting the quadratic term for $PM_{2.5}$ improvement on TICSm trajectory slope ($p$ = 0.04 for the interaction between the quadratic term of $PM_{2.5}$ improvement

and follow-up year, S11 Table), the overall results did not suggest nonlinear effects of AQ improvement on cognitive trajectory slopes (S11 Table).

We found no statistical evidence to suggest that the observed associations substantially differed by age, education, geographic region, ApoE e4 genotypes, or common cardiovascular risk factors after adjusting for multiple comparisons (false discovery rate corrected *p*-values > 0.05 for all tests, Figs 3 and 4).

## Discussion

In this geographically diverse cohort of community-dwelling older women with up to 20 years of follow-up, we found that living in locations with 10-year improvements in ambient AQ in late life was associated with slower cognitive declines. The estimated associations, equivalent to slower declines in general cognitive status observed in women 0.9 to 1.2 years younger or in episodic memory observed in women 1.4 to 1.6 years younger, remained after adjusting for sociodemographics (age; geographic region; race/ethnicity; education; income; employment status; and neighborhood socioeconomic characteristics), lifestyle factors (smoking; alcohol; and physical activity), or clinical characteristics (BMI; depressive symptoms; diabetes; hypercholesterolemia; hypertension; CVD; and hormone therapy). The findings were robust in our sensitivity analyses accounting for various sources of spatiotemporal confounding and were largely unchanged after excluding women with prevalent or incident stroke. Excluding incident dementia cases greatly reduced the associations with slower declines in general cognitive status, but the associations with slower decline in episodic memory were modestly attenuated and remained statistically significant. There was no strong evidence that the slower declines in cognition associated with improved AQ differed by age, education, geographic region, APoE e4 genotype, or cardiovascular risk factors. To the best of our knowledge, this study provides the first epidemiologic evidence supporting the potential benefit of improved AQ on slowing cognitive aging.

To the best of our knowledge, our study adds novel epidemiologic data strengthening the evidence between late-life exposure to ambient air pollution and cognitive decline. Evidence from epidemiological and neurotoxicological studies point to a possible continuum of air pollution neurotoxicity on brain aging. Within the WHIMS suite of studies, we have found that $PM_{2.5}$ is associated with progression of gray matter atrophy in brain areas vulnerable to AD [18,41], episodic memory decline [41,42], and increased risk of clinically significant cognitive impairment [43], demonstrating this continuum in older women. In the ALzheimer and FAmilies cohort, investigators also found that higher exposure to PM was associated with lower cortical thickness in brain regions linked to AD in middle-aged men and women [17]. However, longitudinal epidemiological studies investigating the associations of cognitive declines with $PM_{2.5}$ and $NO_2$ have produced mixed results, including no associations [44–49], significant adverse effects [41,42,50–52], and significant positive associations (suggesting that $PM_{2.5}$ improved cognition function) [53]. In the present study, we found that improved AQ was significantly associated with slower declines in both general cognitive status and episodic memory. Although these findings alone do not prove causality, using a quasi-experimental design, our novel findings along with other supportive evidence from human studies and animal models strengthen the evidence of late-life exposure to ambient air pollution in its contribution to the progression of cognitive aging.

Our study findings have several important public health implications. First, we found that slower cognitive decline was associated with long-term reduction in ambient $PM_{2.5}$ and $NO_2$ levels. In the Children's Health Study, investigators also showed that the benefits of AQ improvement in both $PM_{2.5}$ and $NO_2$ was associated with improved respiratory health [25–

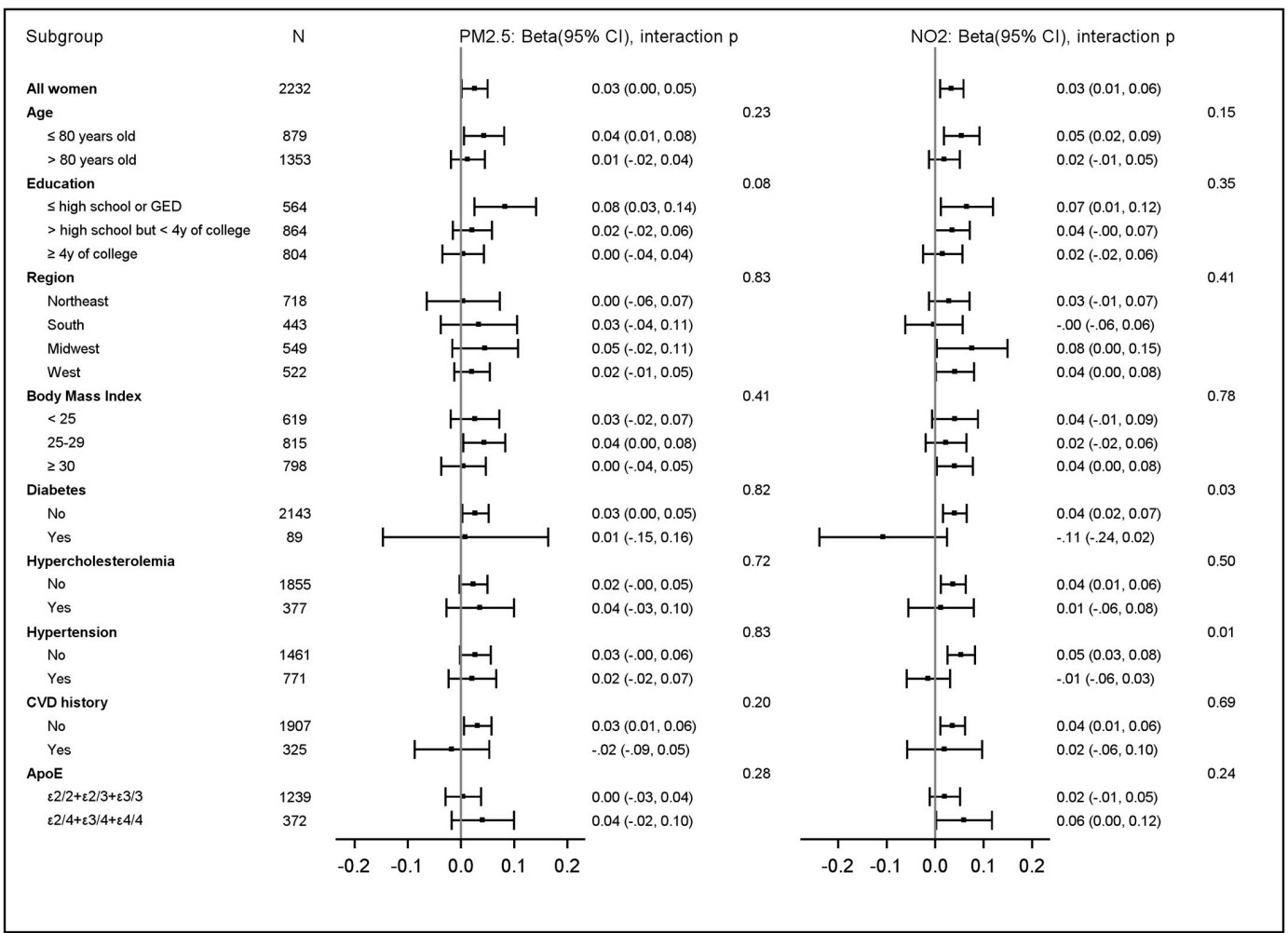

**Fig 3. Estimated associations[a] between AQ improvement[b] and cognitive ability decline[c], stratified by population characteristics.** The bars and whisker represent the regression coefficient beta and corresponding 95% CIs. [a]Association was represented by beta, the regression coefficient estimating the increase in TICSm score per year for each IQR increase of AQ improvement (IQR$_{PM2.5}$ = 1.79 μg/m$^3$; IQR$_{NO2}$ = 3.92 ppb), adjusting for spatial random effect, WHIMS-ECHO enrollment year, age, follow-up year, age interaction with follow-up year, time-varying propensity scores, demographic variables (geographic region and race/ethnicity), socioeconomic factors (education, income, and employment status) and neighborhood characteristics, lifestyle factors (smoking, drinking, and physical activities), prior hormone use, hormone therapy assignment, cardiovascular risk factors (hypertension, diabetes, and hypercholesterolemia), depression, BMI, and CVD histories. [b]Recent exposures were the 3-year average exposures estimated at the WHIMS-ECHO enrollment. Remote exposures were the 3-year average exposures estimated 10 years before the WHIMS-ECHO enrollment. AQ improvement was defined by reduction from remote to recent exposures over the 10-year period. [c]p-Value was calculated using Wald $t$ test for the interaction between AQ improvement and each subgroup unadjusted for multiple comparison. After controlling for multiple comparison using Benjamini–Hochberg approach, false discovery rate corrected p-values > 0.05 for all interaction tests. ApoE, Apolipoprotein E; AQ, air quality; BMI, body mass index; CVD, cardiovascular disease; GED, general educational development; IQR, interquartile range; NO$_2$, nitrogen dioxide; PM$_{2.5}$, fine particulate matter; ppb, parts per billion; TICSm, modified Telephone Interview for Cognitive Status; WHIMS-ECHO, Women's Health Initiative Memory Study-Epidemiology of Cognitive Health Outcomes.

27]. These findings may imply that the observed health benefits of AQ improvement are due to the overall reduced ambient air pollution levels, rather than driven by specific control programs to mitigate either PM$_{2.5}$ or NO$_2$ in the US. Second, although the Clean Air Act mandates that the EPA sets AQ standards that provide a safe margin for susceptible populations [54], our analyses (Fig 3) revealed similar associations comparing subgroups of older women. This observation suggests the resulting benefit of slower cognitive decline associated with AQ improvement in late life may be universal in older women, including those already at greater risk for cognitive decline (e.g., women with high-risk cardiovascular profiles; ApoE e4

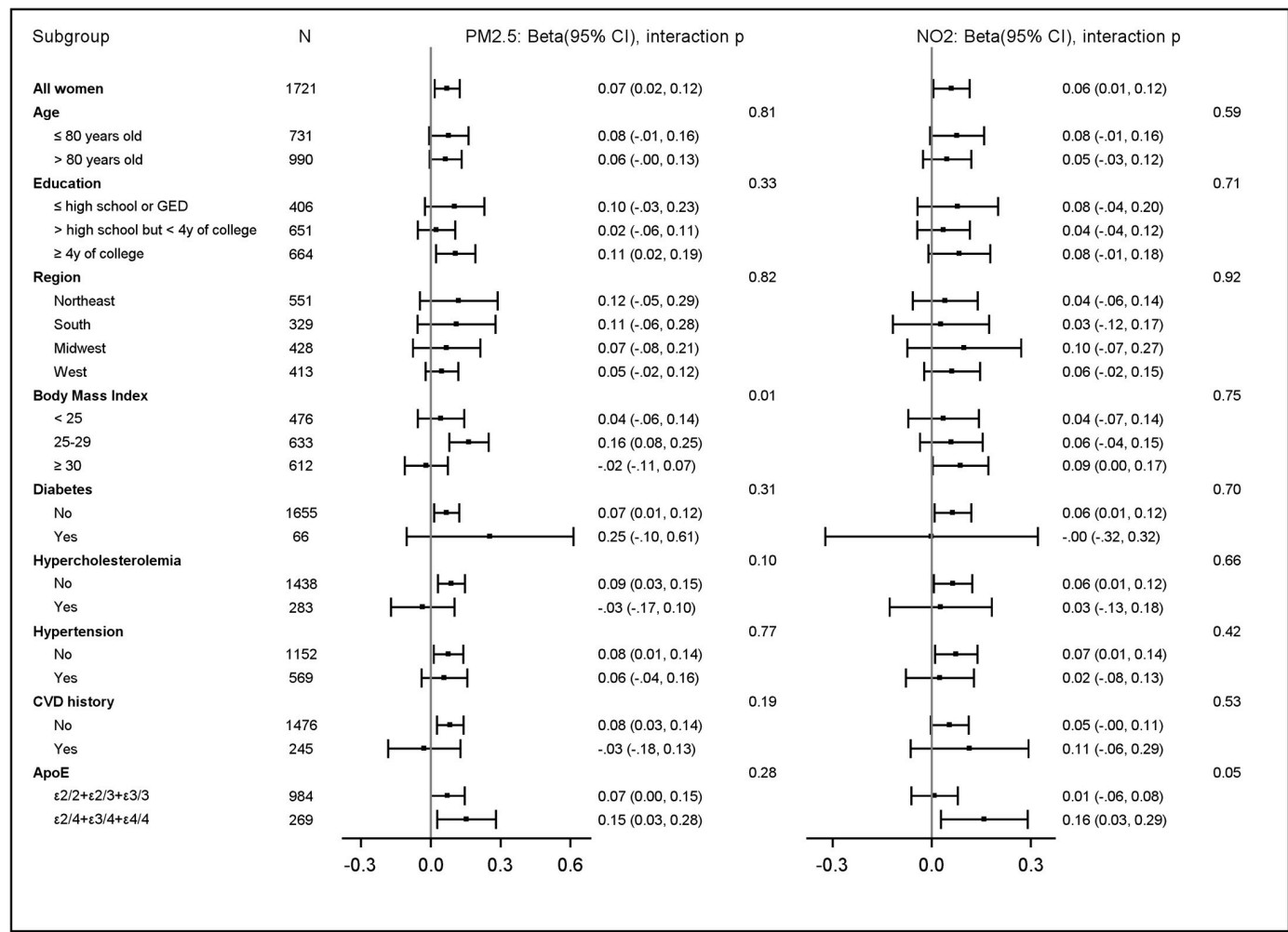

**Fig 4. Estimated associations[a] between AQ improvement[b] and episodic memory decline[c], stratified by population characteristics.** The bars and whisker represent the regression coefficient beta and corresponding 95% CIs. [a]Association was represented by beta, the regression coefficient estimating the increase in CVLT score per year for each IQR increase of AQ improvement ($IQR_{PM2.5} = 1.79$ μg/m$^3$; $IQR_{NO2} = 3.97$ ppb), adjusting for spatial random effect, WHIMS-ECHO enrollment year, age, follow-up year, age interaction with follow-up year, time-varying propensity scores, demographic variables (geographic region and race/ethnicity), socioeconomic factors (education, income, and employment status) and neighborhood characteristics, lifestyle factors (smoking, drinking, and physical activities), prior hormone use, hormone therapy assignment, cardiovascular risk factors (hypertension, diabetes, and hypercholesterolemia), depression, BMI, and CVD histories. [b]Recent exposures were the 3-year average exposures estimated at the WHIMS-ECHO enrollment. Remote exposures were the 3-year average exposures estimated 10 years before the WHIMS-ECHO enrollment. AQ improvement was defined by reduction from remote to recent exposures over the 10-year period. [c]p-Value was calculated using Wald *t* test for the interaction between AQ improvement and each subgroup unadjusted for multiple comparison. After controlling for multiple comparison using Benjamini–Hochberg approach, false discovery rate corrected *p*-values > 0.05 for all interaction tests. ApoE, Apolipoprotein E; AQ, air quality; BMI, body mass index; CVD, cardiovascular disease; CVLT, California Verbal Learning Test; GED, general educational development; IQR, interquartile range; NO$_2$, nitrogen dioxide; PM$_{2.5}$, fine particulate matter; WHIMS-ECHO, Women's Health Initiative Memory Study-Epidemiology of Cognitive Health Outcomes.

carriers). Third, the EPA's projection that the AQ and health benefits achieved as a result of the Clean Air Act Amendments of 1990 are valued at approximately US$2 trillion in 2020 [55] was likely underestimated as it did not include the assessment of brain health, which costs the US economy US$159 to US$215 billion [56]. Fourth, our findings call for future studies to determine whether risk reductions may still be seen with improvements at lower-exposure levels. Increased dementia risks associated with low-level exposures had been reported in Sweden and Canada (range of mean PM$_{2.5}$: 7.6 to 10.4 μg/m$^3$; range of mean NO$_2$: 12.1 to 16.2 ppb) [57–60]. In a subset of WHIMS participants, we previously found that exposure to late-life

$PM_{2.5}$ at levels below the current EPA regulatory standard of 12 μg/m$^3$ was associated with gray matter atrophy in brain areas vulnerable to AD [18]. Therefore, it is imperative to know whether further improvement in low-exposure air pollution levels already in compliance with the current standard still translate to slower cognitive decline. Last, our findings may provide the impetus for future research to consider how AQ improvement could potentially benefit brain development, as studies have found that higher levels of air pollution may hinder cognitive development in children [61,62].

The underlying neurobiological processes driving the observed slower cognitive declines associated with improved AQ are unclear. In our sensitivity analyses excluding women with stroke, the slower cognitive declines with improved AQ were largely unchanged, suggesting that reducing clinical stroke in late life may not make significant contributions to the observed associations. Neuroimaging studies have not shown consistent associations between air pollution and MRI-based measurements of subclinical cerebrovascular disease [9,11,63–65]. By contrast, excluding dementia cases resulted in a substantial attenuation in the estimated associations with TICSm declines, suggesting that the putative benefit of AQ improvement on slowing declines in general cognitive status may be operating by slowing the neuropathological processes near the clinical stage. On the other hand, there was a modest attenuation in the estimated associations with CVLT declines after excluding dementia cases, suggesting the observed associations of slower episodic memory decline with AQ improvement were only partly explained by the underlying neuropathological processes leading to clinical dementia. The remaining association observed among women without dementia implied the possibility of neuroprotective mechanisms underlying the putative benefits on brain health associated with improved AQ. Put together, these results point to the possibility that AQ improvement may benefit the continuum of pathological brain aging. Longitudinal studies with high-quality data on PET scan and fluid-based biomarkers are needed to better understand the underlying neuropathological processes amendable to improved AQ in late life, such as preserving brain volume or maintaining function of neural networks.

We acknowledge several study limitations. First, cognition was assessed using telephone-based interviews rather than the "gold standard" method of face-to-face administration. However, telephone-based cognitive assessment has been shown to be both reliable and valid and may improve study validity by increasing retention and data completeness [66]. Second, we estimated the ambient levels of air pollution at residential locations and detailed information on time-activity patterns were not available. However, the use of individual-level estimates of reduced exposures to ambient air pollutants is appropriate for studying the public health benefits [23,25–27] since these pollutants are regulated by the EPA. Third, the errors in predicting the air pollution estimates may have varied throughout the 10-year period and may have contributed uncertainty to our analyses. Fourth, we could not completely rule out unmeasured confounding by other environmental factors (e.g., noise and green space) with longitudinal changes that may be concurrent with improved AQ. However, noise levels have been increasing [67], and green space has been decreasing over time due to increasing urbanization [68], so it is unlikely they would contribute to the health benefits of long-term AQ improvement. Fifth, regression to the mean may be a concern due to measuring changes in AQ across only 2 time periods, which may have attenuated the estimated associations between AQ improvement and cognitive decline, as suggested by the results of our sensitivity analyses further adjusting for either recent or remote exposures (S7 Table). Sixth, because our study sample came from a nonrandom selective process (Fig 1), we could not completely rule out the possibility of selection biases. Last, our findings cannot be generalizable to men or younger women.

Our study had several major methodological strengths. First, the geographic diversity coupled with the extended longitudinal follow-up provided an ideal environmental context to

explore the health benefits of AQ improvement across the US. Second, the estimated benefits were based on within-cohort comparisons, instead of cross-cohort comparisons done in previous studies [25–27], greatly reducing possible confounding stemming from between-cohort differences [69]. Third, using individual-level estimates of ambient air pollution to define improved AQ further reduces the spatial confounding arising from using county-/city-/community-level estimates as was done in previous studies [23–27]. Last, our analyses also accounted for different sources of spatiotemporal confounding, such as adjustment of WHIMS-ECHO enrollment year, including a random effect of clinic centers, and further adjusting for temporal changes in relevant covariates that may correlate with AQ improvement.

In conclusion, we found that long-term AQ improvement during late life was robustly associated with slower rates of cognitive declines among older women. The associations were similar for both $PM_{2.5}$ and $NO_2$ and did not differ by age, geographic region, education, or cardiovascular risk factors. Future studies are needed to improve our understanding on the underlying neuropathological processes that may be amendable by reducing exposure to late-life ambient air pollution.

## Supporting information

**S1 Checklist. STROBE Statement—Checklist of items that should be included in reports of cohort studies.** STROBE, Strengthening the Reporting of Observational Studies in Epidemiology.
(DOCX)

**S1 Text. Covariates assessed at WHI inception.** WHI, Women's Health Initiative.
(DOCX)

**S2 Text. Assessment of US Census tract-level socioeconomic characteristics of residential neighborhood.**
(DOCX)

**S3 Text. Assessment of covariates at the WHIMS-ECHO enrollment.** WHIMS-ECHO, Women's Health Initiative Memory Study-Epidemiology of Cognitive Health Outcomes.
(DOCX)

**S4 Text. Assessment of longitudinal changes in covariates from the WHI inception to WHIMS-ECHO enrollment.** WHI, Women's Health Initiative; WHIMS-ECHO, Women's Health Initiative Memory Study-Epidemiology of Cognitive Health Outcomes.
(DOCX)

**S5 Text. APoE genotype data.** APoE, Apolipoprotein E.
(DOCX)

**S6 Text. Equations of 3-level linear mixed effect models.**
(DOCX)

**S7 Text. Time-varying propensity score approach to adjust for selective attrition due to loss to follow-up.**
(DOCX)

**S8 Text. Multiple imputation.**
(DOCX)

**S9 Text. Ascertainment of probable dementia.**
(DOCX)

**S1 Fig. Map of US with region, state, and clinic sites of WHIMS participants.** The direct link to the base layer of the map used in this figure: https://services.arcgis.com/P3ePLMYs2RVChkJx/arcgis/rest/services/USA_States_Generalized/FeatureServer. Maps were created using ArcGIS software by Esri. ArcGIS and ArcMap are the intellectual property of Esri and are used herein under license. Copyright Esri. All rights reserved. For more information about Esri software, please visit www.esri.com. WHIMS, Women's Health Initiative Memory Study.
(TIFF)

**S1 Table. Comparing study samples with repeated CVLT measures versus excluded due to no repeated CVLT measures.** CVLT, California Verbal Learning Test.
(DOCX)

**S2 Table. Pearson correlations between AQ measures.** AQ, air quality.
(DOCX)

**S3 Table. Distribution of cognitive test scores by visit.**
(DOCX)

**S4 Table. Distribution of cognitive scores at baseline and last visits by population characteristics.**
(DOCX)

**S5 Table. Summary of sensitivity analyses for the associations between AQ improvements and cognitive decline, with adjustment of covariates assessed at the WHIMS-ECHO enrollment or Changes from WHI inception to WHIMS-ECHO enrollment.** AQ, air quality; WHI, Women's Health Initiative; WHIMS-ECHO, Women's Health Initiative Memory Study-Epidemiology of Cognitive Health Outcomes.
(DOCX)

**S6 Table. Summary of the associations between AQ improvements and cognitive decline, with missing AQ measures and covariates imputed using multiple imputation.** AQ, air quality.
(DOCX)

**S7 Table. Summary of the associations between AQ measures and cognitive decline, with single exposure or multiple exposures in one model.** AQ, air quality.
(DOCX)

**S8 Table. Summary of the associations between AQ improvements and cognitive decline, excluding women with dementia or stroke.** AQ, air quality.
(DOCX)

**S9 Table. Summary of the associations between AQ improvement and quadratic term of cognitive decline.** AQ, air quality.
(DOCX)

**S10 Table. Summary of the associations between AQ improvement and cognitive declines, with the nonlinear change in cognitive trajectories.** AQ, air quality.
(DOCX)

**S11 Table. Summary of the associations with cognitive trajectory slopes, evaluating nonlinear associations with AQ improvement.** AQ, air quality.
(DOCX)

## Author Contributions

**Conceptualization:** Diana Younan, Xinhui Wang, Helena C. Chui, Margaret Gatz, Stephen R. Rapp, Jiu-Chiuan Chen.

**Data curation:** Xinhui Wang, Daniel P. Beavers, Joel D. Kaufman, Gregory A. Wellenius, Eric A. Whitsel.

**Formal analysis:** Xinhui Wang, Joshua Millstein, Andrew J. Petkus.

**Funding acquisition:** Joel D. Kaufman, Jiu-Chiuan Chen.

**Investigation:** Mark A. Espeland, Helena C. Chui, Susan M. Resnick, Gregory A. Wellenius, JoAnn E. Manson, Stephen R. Rapp.

**Methodology:** Xinhui Wang, Joshua Millstein, Andrew J. Petkus, Daniel P. Beavers, Mark A. Espeland, Jiu-Chiuan Chen.

**Project administration:** Jiu-Chiuan Chen.

**Software:** Xinhui Wang.

**Supervision:** Jiu-Chiuan Chen.

**Validation:** Xinhui Wang.

**Visualization:** Diana Younan, Xinhui Wang.

**Writing – original draft:** Diana Younan, Xinhui Wang, Jiu-Chiuan Chen.

**Writing – review & editing:** Diana Younan, Xinhui Wang, Joshua Millstein, Andrew J. Petkus, Daniel P. Beavers, Mark A. Espeland, Helena C. Chui, Susan M. Resnick, Margaret Gatz, Joel D. Kaufman, Gregory A. Wellenius, Eric A. Whitsel, JoAnn E. Manson, Stephen R. Rapp, Jiu-Chiuan Chen.

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
