## [Editor Report · Decision Letter 0]

26 Feb 2021

Dear Dr Younan, 

Thank you for submitting your manuscript entitled "Association of Air Quality Improvement with Slower Decline of Cognitive Function in Older Women" for consideration by PLOS Medicine.

Your manuscript has now been evaluated by the PLOS Medicine editorial staff as well as by an academic editor with relevant expertise and I am writing to let you know that we would like to send your submission out for external peer review.

Kind regards,

Dr Raffaella Bosurgi

Executive Editor 

PLOS Medicine

---

## [Decision Letter · Decision Letter 1]

27 Aug 2021

Dear Dr. Younan,

Thank you very much for submitting your manuscript "Association of Air Quality Improvement with Slower Decline of Cognitive Function in Older Women" (PMEDICINE-D-21-00961R1) for consideration at PLOS Medicine. 

Your paper was evaluated by three independent reviewers, including a statistical reviewer, and was discussed among all the editors here and with an Academic Editor with relevant expertise. The reviews are appended at the bottom of this email and any accompanying reviewer attachments can be seen via the link below:

[LINK]

In light of these reviews, I am afraid that we will not be able to accept the manuscript for publication in the journal in its current form, but we would like to consider a revised version that addresses the reviewers' and editors' comments. Obviously we cannot make any decision about publication until we have seen the revised manuscript and your response, and we plan to seek re-review by one or more of the reviewers. 

We expect to receive your revised manuscript by Sep 17 2021 11:59PM. Please email us (plosmedicine@plos.org) if you have any questions or concerns.

We look forward to receiving your revised manuscript. 

Sincerely,

Louise Gaynor-Brook, MBBS PhD

Associate Editor 

PLOS Medicine

plosmedicine.org

Comments from the Academic Editor:

This is a very good attempt at trying to relate improvement in environmental pollution with cognitive decline. They have tried to control demographic, lifestyle and clinical factors. It is very likely that not all confounding factors can be accounted for in such a study. The question that comes up is, do the people living in an environment that is improving differ systematically from those who live in an environment that is not? One can account for edu and income, but is that enough? Do they have a generally healthy lifestyle? Do they exercise more? Do they eat healthily? Some of these are difficult to control for?

The other issue is that the improvement in the environment occurred over the 10 years prior to the recruitment. Has that improvement continued through the study? Did other areas catch up?

General comments:

Throughout the paper, please adapt reference call-outs to the following style: "... Alzheimer’s disease (AD) neuropathologies [17,18].” (noting the absence of spaces within the square brackets).

Please avoid using ‘average’ - please specify.

Please replace "subject" with participant, patient, individual, or person.

Thank you for providing a data availability statement. PLOS Medicine requires that the de-identified data underlying the specific results in a published article be made available, without restrictions on access, in a public repository or as Supporting Information at the time of article publication, provided it is legal and ethical to do so. If the data will be freely available upon request, please state the owner of the data set and specific contact information for data requests (web or email address). Please note that a study author cannot be the contact person for the data.

Title: Please revise your title according to PLOS Medicine's style. Your title must be nondeclarative. Please place the study design in the subtitle (ie, after a colon). We suggest “Association between air quality improvement and changes in cognitive function in community-dwelling older women: A longitudinal cohort study“

Abstract:

Abstract Background: Please revise to “Air quality (AQ) has been associated with…”, and please temper assertions of primacy by adding “to the best of our knowledge” or similar. The final sentence should clearly state the study question.

Abstract Methods and Findings:

Please provide brief demographic details of the study population (e.g. sex, age, ethnicity, etc)

Please clarify what is represented by the units used for the numbers presented in the abstract e.g. (βPM2.5=0.026/year per IQR=1.79 µg/m3). It may be easier to describe this.

Please revise the sentence beginning “ Individual-level improved AQ calculated…”

Line 57 - Please make clear that the estimates were 3-year averages

Please specify the important dependent variables that are adjusted for in the analyses.

Please clarify whether the results presented are adjusted analyses, and please specify the comparison group.

Please define ppb at first use 

In the last sentence of the Abstract Methods and Findings section, please describe 2-3 of the main limitation(s) of the study's methodology."

Abstract Conclusions:

Please begin your Abstract Conclusions with "In this study, we observed ..." or similar, to summarize the main findings from your study, and expand a little on the implications of your study without overstating your conclusions.

Author Summary:

In the final bullet point of ‘What Do These Findings Mean?’, please describe the main limitations of the study in non-technical language.

Introduction:

Please indicate whether your study is novel and how you determined that (being careful to temper assertions of primacy by adding “to the best of our knowledge” or similar). If there has been a systematic review of the evidence related to your study, please refer to and reference that review and indicate whether it supports the need for your study. 

Methods:

Did your study have a prospective protocol or analysis plan? Please state this (either way) early in the Methods section. If a prospective analysis plan (from your funding proposal, IRB or other ethics committee submission, study protocol, or other planning document written before analyzing the data) was used in designing the study, please include the relevant prospectively written document with your revised manuscript as a Supporting Information file to be published alongside your study, and cite it in the Methods section. A legend for this file should be included at the end of your manuscript. If no such document exists, please make sure that the Methods section transparently describes when analyses were planned, and if/when reported analyses differed from those that were planned. Changes in the analysis-- including those made in response to peer review comments-- should be identified as such in the Methods section of the paper, with rationale. If a reported analysis was performed based on an interesting but unanticipated pattern in the data, please be clear that the analysis was data-driven.

Thank you for providing your STROBE guideline. Please add the following statement, or similar, to the Methods: "This study is reported as per the Strengthening the Reporting of Observational Studies in Epidemiology (STROBE) guideline (S1 Checklist)." When completing the checklist, please use section and paragraph numbers, rather than page numbers which will likely no longer correspond to the appropriate sections after copy-editing.

Results: 

Please incorporate S1 Table into your Results section as Table 1, showing the baseline characteristics of the study population.

Line 214 - please clarify what is represented by ‘±2.7’ (presumably SD)

Line 230 - please be careful to avoid causative language; please revise sentence beginning “Older women residing…” to “Residing in locations with greater AQ improvement was associated with slower rates of decline... “ or similar 

Line 232 - please revise sentence beginning “For each IQR increment of improved AQ…” to “Each IQR increment … was associated with…“ or similar 

Discussion:

Line 347 - please temper assertions of primacy by adding “to the best of our knowledge” or similar. Please also revise the use of ‘causal’ on lines 347 and 362; we suggest ‘strengthen the evidence' or similar.

Figures:

Please consider avoiding the use of red and green together in order to make your figure more accessible to those with colour blindness (e.g. Fig 2).

Please indicate in the figure caption the meaning of the bars and whiskers in Figure 3.

Please define all abbreviations used in the figure legend of each figure.

Tables:

Please define all abbreviations used in the table legend of each table.

When a p value is given, please specify the statistical test used to determine it in the table legend.

Table 2, S3 Table - Please also provide the unadjusted analyses/models.

S5 Table - please clarify how the models differ in the table legend.

References:

Please ensure that journal name abbreviations match those found in the National Center for Biotechnology Information (NCBI) databases, and are appropriately formatted and capitalised.

Please also see https://journals.plos.org/plosmedicine/s/submission-guidelines#loc-references for further details on reference formatting. 

Comments from the reviewers:

Reviewer #1: This study attempts to leverage a nationwide cohort of community dwelling older women with individual-level air pollution exposure estimates (1996-2012) and annual assessments of late-life cognitive function (2008−2018). The authors hypothesise that improved AQ, as indicated by reductions in PM2.5 and NO2, is associated with slower rate of cognitive decline in older women. 

Comments:

The STROBE checklist has been suitably provided in the supplementary material.

"We conducted a longitudinal study on a geographically diverse cohort of community dwelling older women (N=2880; aged 74-92) enrolled in the Women's Health Initiative (WHI) Memory Study (WHIMS)-Epidemiology of Cognitive Health Outcomes (WHIMS-ECHO) study ... This resulted in a final analytic sample of 2232 women for the analyses on general cognitive status, a subset of which (n=1721) was used for the analyses on episodic memory (Fig 1A)."

Can the authors please comment on whether they believe this cohort to be representative of the wider population? I.E. Are there any potential causes of bias that should be highlighted?

"WHIMS-ECHO participants received annual neuropsychological assessments via centralized telephone-administered cognitive interviews conducted by trained and certified staff."

Can the authors please discuss the potential for self-reporting bias in the study limitations?

It is noted that, with respect to covariates: "Good reliability and validity of both the self-reported medical histories and the physical measures have been documented (37-39)."

"Participants were asked 16 questions with items assessing several cognitive functions. The TICSm score (0-50) was defined as the total number of correct responses, with higher scores indicating better cognitive functioning."

Can these questions and scoring computation please be provided in the supplementary material?

"In the present study, we used all longitudinal data collected from telephone based assessments until June 2018 (29)."

Can this be extended to 2020?

"We used regionalized universal kriging models to estimate annual concentrations (1996-2012) of fine particulate matter (PM2.5) and nitrogen dioxide (NO2) at residential locations. Estimates were averaged over 3-year periods immediately preceding (recent exposure) and 10 years prior to (remote exposure) WHIMS ECHO enrollment."

Did the authors consider including a more current AQ assessment, i.e. after enrolment, as well as these two time points? 

"We used linear mixed effect models to examine whether AQ improvement before WHIMS-ECHO enrollment was associated with average decline rates in the TICSm and CVLT trajectories during the follow-up".

This is a technically appropriate methodology, assuming linearity holds. 

Overall, this is a clearly written article, with excellent communication of information via figures.

The authors have included an extensive array of potential confounders within the model, and completed a thorough set of sensitivity analyses demonstrating the robustness of the study outcomes.

Furthermore, the study limitations have been acknowledged suitably within the discussion section of the manuscript.

Reviewer #2: The study of Younan et al examines whether air pollution improvement of NO2 and PM2.5 over a 10-years period, is associated with cognitive improvement in elderly women from US. They used data from 2,000 women enrolled in the Women's Health Initiative Memory Study-Epidemiology of Cognitive Health Outcomes who provided on general cognitive status and episodic memory decline. Authors observed that greater air quality improvement for NO2 and PM2.5 was associated with slower decline in both outcomes. The study is well-designed and well-written. Main strengths include the repeated outcomes measurements and the large sample size. Hereby I include some comments:

Major comments:

- Indoor air quality is not considered in the present study and this may have a great impact in the association since elderly women spent most of the time indoors. 

- Authors are aware that they are not considering other features of the urban environment such as green spaces nor noise. It would be nice to better explain why the authors think they do not have affected the associations observed (also refer to some papers that have seen an association between noise and dementia such as Yu et al Epidemiology. 2020 Nov 1;31(6):771-778). I suggest to draw a digital acyclic graph (DAG) including noise, green spaces, and other features of the urban environment to better understand their potential contribution to the studied associations. 

- I am missing a table with a description of the outcomes and the number of interviews performed for each woman (I think this is not mentioned in the manuscript; also report the mean and SD in the text of the number of interviews). It would be also informative to show the distribution of each test in each interview. Have the authors checked whether the number of interviews affected the association? (major number of interviews � stronger associations?). A table including the scores at baseline and the scores in the last interview would be also an asset and comparing this (or the distribution of the outcomes) with women's characteristics (age, BMI, education etc). 

- I would include the results of the imputed dataset as main results and move the complete case ones to the supplementary material. 

- It is not clear to me how the authors have selected the confounders since some of them are not associated with air quality improvement as shown in Table 1 (if the based on the association with exposure and outcome) and some can act as potential mediators (e.g. physical activity can mediate the association between urban design - air pollution - physical activity - dementia; cardiovascular health since some of the indicators such as obesity have been associated with air pollution exposure; or mental health). The DAG can help in visualizing which covariates act as confounders and which one can act as potential mediators or effect modifiers.

- In figure 3B it seems that BMI modifies the association between air quality but this is not mentioned. 

- Clinical sites are firstly mentioned in the statistical analysis. A good description of them should be included in the study population and also in Table 1 as another population characteristic. 

- I would include the association between remote and recent NO2 and PM2.5 exposure with each of the outcomes as main analysis and then report the air quality improvement in the same table similar to Table S5 (very difficult to understand what each Models mean). 

- Please, discuss how the estimation of air pollution can have varied throughout the 10 years period and how this may have affected the results (and perform some analyses if needed). 

Minor comments

- Introduction and discussion - it would be nice to discuss some equivalent analysis between air pollution and cognition development in children such as the study conducted in Barcelona schools (PLoS Med. 2015 Mar 3;12(3):e1001792).

- How deaths are treated in the analysis?

- Use the same wording in the text and in Figure 1 regarding the outcomes assessed. 

- I would not say 'Older women' in sentences 216 and 230 because it seems the authors are comparing women within the study based on their age. I would just say 'Women'. 

- Include the confounding variables in the models in Figure 2 footnote. 

- In table 2, I would change Model I and Model II for Minimally adjusted model and Fully adjusted model. If authors select confounders based on the DAG, then the minimally adjusted model is no longer needed. 

Reviewer #3: This article reports a detailed evaluation of whether improvement in air quality is associated with cognitive change in older women using the Women's Health Initiative study dataset. The authors have linked this to geocoded measures of environment and air quality over time. Air pollution is a risk factor for cognitive decline and dementia, so this is an important study because it evaluates whether risk reversal is associated with any cognitive benefit. The investigation is rigorous and well considered. Authors are careful not to claim causality as the study is observational and it is possible that there are unmeasured confounders. There are some areas of the methodology that were unclear such as:

1. It is unclear how exposure to air pollution over time was estimated at the level of the individual participant, particularly when they moved residence during the study. Can the authors please explain in sufficient detail for full replication of methods, exactly how an individual's 'air pollution reduction' score was derived?

2. What is the validity and reliability of the environmental measures and air pollution measures that were used in this study? 

3. Did the authors take into consideration where participants spent their time during each day or did they only examine residential location of participants? for example, an individual may live in one area but work in another and effectively spend more time away from their residential location. 

4. When participants moved, how was time treated in the statistical modelling. i.e what was the unit of time - was it days, weeks, months or a year?

5. The geographical regions would not be well understood outside of the USA. it is suggested that these are better described in the method or supplement. Similarly the education levels are not described in widely understood terms. In the US 'College' refers to what other countries call 'University' or 'tertiary' level education. To ensure wider relevance it would be useful to define this variable more clearly. 

6. More information is needed on the APOE genotyping and how this variable was classified and treated in statistical analyses as there are different practices in the literature eg. some studies exclude certain combinations of e2, e3, and e4 alleles or combine e4 homozygous and e4 heterozygous. APOE e4 status should also be reported in the descriptive statistics.

7. Can the authors provide any information on how the missing data on the CVLT would have biased results?

8. Were these the only two cognitive measures collected by the WHIMS study? Other papers mention the 3MSE which is an extended Mini-Mental State Examination.

[LINK]

---

## [Decision Letter · Decision Letter 2]

18 Nov 2021

Dear Dr. Younan,

Thank you very much for re-submitting your manuscript "Association between air quality improvement and slower cognitive decline in community-dwelling older women: A longitudinal cohort study" (PMEDICINE-D-21-00961R2) for review by PLOS Medicine.

I have discussed the paper with my colleagues and the academic editor and it was also seen again by two reviewers. I am pleased to say that provided the remaining editorial and production issues are dealt with we are planning to accept the paper for publication in the journal.

[LINK]

We look forward to receiving the revised manuscript by Nov 25 2021 11:59PM.   

Sincerely,

Callam Davidson (on behalf of Louise Gaynor-Brook)

Associate Editor 

PLOS Medicine

plosmedicine.org

Requests from Editors:

Your title should be nondeclarative. Please update to ‘Air quality and cognitive decline in community-dwelling older women in the United States: A longitudinal cohort study’.

The Data Availability Statement (DAS) requires revision. If the data are owned by a third party but freely available upon request, please note this and state the owner of the data set and contact information for data requests (web or email address). The URL you have provided in your statement appears to be a broken link. 

Line 53: Please update to ‘in older women aged 74-92 years’.

Please include basic information about the nationwide remit of the WHIMS-ECHO cohort in your abstract methods and findings (i.e. ‘Participants resided in the 48 contiguous U.S. states and were recruited from more than 40 study sites located in 24 states and Washington, D.C.’).

Your Author Summary should not recycle text from the abstract as far as is possible, please revisit and ensure you have not copied text directly across (e.g. bullet point 1).

Please consider relocating some of the text describing your study population from the methods to the results section. The methods section ought to describe the WHIMS study and define inclusion/exclusion criteria, while the final analytic sample and its demographic characteristics would be better placed in the results. 

Your rebuttal letter notes that you updated some of your analyses based on reviewer comments. Changes in the analysis, including those made in response to peer review comments, should be identified as such in the Methods section of the paper, with rationale (this can be included in the Study design and population subsection).

Line 360: Ensure footnote ‘f’ begins on a new line.

Line 467: Please update ‘studies’ to ‘at least one study’ as you only cite one study.

Please remove your ‘Financial disclosure’, ‘Competing interests’, and ‘Data statement’ from the end of the main text; all of this information will be captured as metadata based on your responses to the submission form questions (so please ensure all relevant information is included in your answers).

Similarly, the ‘Ethical approval’ statement (line 545) can be removed as this is already included in your Methods section.

Please check the formatting of reference 50.

You have included a map in your updated supplementary materials. Please confirm that the appropriate usage rights apply to the use of this map. Please see our guidelines for map images: https://journals.plos.org/plosmedicine/s/figures#loc-maps

Comments from Reviewers:

Reviewer #1: The authors have satisfactorily considered and responded to each comment in turn, providing clarifications and amending the manuscript where required.

Reviewer #2: The authors properly replied to all my concerns. I do not have any other additional comment.

[LINK]

---

## [Editor Report · Decision Letter 3]

15 Dec 2021

Dear Dr Younan, 

On behalf of my colleagues and the Academic Editor, Prof. Perminder Sachdev, I am pleased to inform you that we have agreed to publish your manuscript "Air quality improvement and cognitive decline in community-dwelling older women in the United States: A longitudinal cohort study" (PMEDICINE-D-21-00961R3) in PLOS Medicine.

Before your manuscript can be formally accepted you will need to complete some formatting changes, which you will receive in a follow up email. Please be aware that it may take several days for you to receive this email; during this time no action is required by you. Once you have received these formatting requests, please note that your manuscript will not be scheduled for publication until you have made the required changes. Given our busy publication schedule, we are planning to publish your paper in early February 2022 (the exact date will be communicated to you once confirmed). 

PRESS

Sincerely, 

Louise Gaynor-Brook, MBBS PhD 

Associate Editor, PLOS Medicine